

# A history-matching analysis of the Antarctic Ice Sheet since the last inter-glacial – Part 1: Ice sheet evolution

Benoit S. Lecavalier[1], Lev Tarasov[1]

[1]Department of Physics and Physical Oceanography, Memorial University of Newfoundland, St. John's, Canada

Correspondence to: Benoit S. Lecavalier (b.lecavalier@mun.ca)

**Abstract.** In this study we present the evolution of the Antarctic Ice Sheet (AIS) since the last interglacial. This is achieved by
means of a history-matching analysis where a newly updated observational database (AntICE2, Lecavalier et al., 2023) was used
to constrain a large ensemble of 9,293 model simulations. The Glacial Systems Model (GSM) configured with 38 ensemble parameters was history matched against observations of past ice extent, past ice thickness, past sea level, ice core borehole temperature profiles, present-day uplift rates, and present-day ice sheet geometry and surface velocity. Successive ensembles were
used to train Bayesian Artificial Neural Network emulators. The parameter space was efficiently explored to identify the most
relevant portions of the parameter space through Markov Chain Monte Carlo sampling with the emulators. The history matching
ruled out model simulations which were inconsistent with the observational constraint database.
During the Last Interglaciation (LIG), the AIS yielded several meters equivalent sea-level (mESL) grounded ice volume deficit relative to present with subsurface ocean warming during this period being the key uncertainty. At the global Last Glacial Maximum
(LGM), the best-fitting sub-ensemble of AIS simulations reached an excess grounded ice volume relative to present of 9.2 to 26.5
mESL. Considering the data does not rule out simulations with an LGM grounded ice volume > 20 mESL with respect to present,
the AIS volume at the LGM can partly explain the missing ice problem and help close the LGM sea-level budget. Moreover, during
the deglaciation, the state space estimation of the AIS based on the GSM and near-field observational constraints allow only a
negligible Antarctic Melt Water Pulse 1a contribution (-0.2 to 0.3 mESL).

## 1. Introduction

The Antarctic Ice Sheet (AIS) has been identified as a major source of uncertainty to future sea level change (Meredith et al.,
2019; Masson-Delmotte et al., 2021). It is one of the slowest components of the climate system given that its interior responds
on 100 kyr timescales. Therefore, studying the past evolution of the AIS can quantify the sensitivity of the ice sheet to past warm
and cold periods, and facilitate the interpretation and projection of contemporary and future ice sheet changes and
corresponding sea level rise. This is primarily achieved using model simulations that aim to reconstruct past changes of the AIS
(Golledge et al., 2012; DeConto and Pollard, 2016a; Albrecht et al., 2020b). However, relevant modelling studies to date are
generally characterized by limited parameter sampling, reliance on hand-tuning, incomplete validation against observational
constraints, and the absence of meaningful uncertainty analysis. As such, the relationship of the resultant simulations to the
actual past ice sheet evolution is unclear. This is particularly relevant given ice sheet instabilities could potentially contribute
metres to sea-level rise over the next two centuries (Rignot et al., 2014; DeConto and Pollard, 2016b; Pattyn and Morlighem,
2020; Edwards et al., 2019a).
Model deficiencies are categorized as follows: approximations of the relevant dynamical equations, missing physics, unresolved
subgrid processes, limited model resolution, and boundary and initial condition uncertainties. The variation of model parameters
is generally the primary (and to date usually the only) method to represent the bulk of the uncertainties associated with these
model limitations. The model ensemble parameters form a potentially high-dimensional parameter space from which a sample
of each individual ensemble parameter, termed a parameter vector, represents one simulation. Previous modelling studies have
generally conducted a limited exploration of the parameter space, generally using less than six ensemble parameters (Denton
and Hughes, 2002; Huybrechts, 2002; Pollard and DeConto, 2009a; Golledge et al., 2014a; Pollard et al., 2016; DeConto and
Pollard, 2016b), and even fewer studies have incorporated the available field observations to constrain their models (Golledge
et al., 2012; Whitehouse et al., 2012a; Albrecht et al., 2020a, b). A large ensemble analysis exceeding thousands of simulations,
supplemented by machine learning emulation has been effectively conducted to explore North American Quaternary ice sheets
(Tarasov et al., 2012) but has yet to be applied to the AIS.
In this study, we present a large ensemble of simulations on the last glacial cycle evolution of the AIS with a high degree of
confidence that it approximately brackets the true AIS history (subject to some explicit caveats presented in the conclusions).
The resultant approximate history-matching analysis explores several fundamental questions about the AIS. The main research
questions answered in this study are: the AIS sea-level contribution during the Last Interglacial (LIG) at ca. 125 ka and Melt Water
Pulse 1a (MWP-1a) around 14.6 ka; the temporal and volume changes of the AIS around the Last Glacial Maximum (LGM; ca. 19-



26 ka); and the influence of past uncertainties on the present-day (PD) AIS. Antarctic glacial isostatic adjustment (GIA) evolution and relative sea-level change are examined in an accompanying paper (Lecavalier et al., 2024).



**Figure 1:** a) Antarctic continent and names of locations mentioned in the study are shown alongside the Antarctic ICe sheet Evolution database version 2 (AntICE2) database (symbols), the main Antarctic sectors delineated by the dark red outlines, and key cross section profiles (orange lines). The data ID numbers and ice core names are labelled in Figure S1. The Antarctic basemap was generated using Quantarctica (Matsuoka et al., 2021).

Our understanding of the AIS has dramatically increased over the past several decades through remote sensing and field campaigns. A large portion of AIS research and resources evaluate the PD state, and the processes and drivers of contemporary changes. Too often, past and future AIS simulations solely rely on the PD ice sheet geometry and surface velocity to constrain and initialize their models (Martin et al., 2019). This fails to recognize that the contemporary AIS is not in a steady state and disregards the past trajectory of the ice sheet. To address the latter, it is important to incorporate valuable albeit limited paleo
observations to constrain and initialize AIS simulations. Nonetheless, our knowledge of the PD AIS state represents our most





powerful constraints and well-defined boundary conditions. For reference, an Antarctic map with places named in the paper is given in Fig. 1.

Large sections of the AIS are marine-based (Fig. 1) and are susceptible to marine ice sheet instabilities (MISI) and potentially marine ice cliff instabilities (MICI) that could contribute a mESL by the end of the century (Golledge et al., 2015a; DeConto and
Pollard, 2016a; Edwards et al., 2019b). The PD mass balance of the AIS has been inferred using a variety of methods which have in turn identified the Amundsen Sea sector of the West Antarctic Ice Sheet (WAIS) as a major contributor to the negative mass balance of the AIS (Shepherd et al., 2018). However, a common requirement across geodetic mass balance inferences of the AIS is the background viscous GIA signal which represents a major source of uncertainty (Whitehouse et al., 2019). The AIS mass balance from 1992 to 2017 was -109 ± 56 Gt/yr (7.6 ± 3.9 mm of sea level rise) (Shepherd et al., 2018). These estimates use poorly
constrained GIA estimates that are based on a limited exploration of uncertainties against observational constraints (Otosaka et al., 2023).

There remain several outstanding research questions regarding the past evolution of the AIS that revolve around the sensitivity and susceptibility of the AIS to past and future climate change. In this study we primarily focus on those pertaining to the grounded ice volume of the AIS since the LIG. A history-matching analysis requires observational data to initialize, force,
constrain, and score model simulations. Moreover, this needs clearly defined observational uncertainties, quantified internal model discrepancies, and reasonable external discrepancy estimations. The robustness of the history-matching analysis results are contingent on the completeness of the error model and an adequate exploration of the parameter phase-space. Given the system nonlinearities, as well as data and model uncertainties, it is highly unlikely that any single model simulation will actually closely replicate past ice sheet evolution. As such, a much more reasonable objective is to produce an envelope of model
reconstructions that convincingly bracket the true evolution, thus confidently bounding the trajectory of the actual system. The history-matching analysis produces bounds of the AIS evolution which improve our understanding of the sea-level budget during key periods of interest: LIG, LGM, and MWPs (Fig. 2). Another product of the history-matching analysis is an ensemble of AIS reconstructions consistent with observational constraints which can be applied as orographic boundary conditions and/or freshwater forcing in general circulation models to better understand atmosphere-ocean circulation and $CO_2$ outgassing in the
past.

Proxy data are required to force and constrain paleo ice sheet and climate simulations. These efforts have increased our understanding of processes, triggers, and feedbacks of past climate change (Lemieux-Dudon et al., 2010; Shakun et al., 2012; Rasmussen et al., 2014). In this study an unprecedented quantity of data and computational resources are used to reconstruct the evolution of the AIS. The observational constraint data is from the new Antarctic ICe sheet Evolution database version 2 (AntICE2,
Lecavalier et al., 2023). Moreover, the model uses a variety of ice core data including the EPICA Dome C (EDC) ice core water isotope record, a proxy for Antarctic air temperature (EPICA, 2004; Jouzel et al., 2007). Key periods of interest referred in the text are labeled alongside the EDC record in Fig. 2.

The LIG is a warm period (129-116 ka; MIS 5e) with global mean temperatures inferred to be 0.5 to 1.0 ºC warmer than preindustrial (Turney et al., 2020; Fischer et al., 2018;(Hoffman et al., 2017), with even warmer amplified polar temperatures (Otto-
Bliesner et al., 2021; Yau et al., 2016). Moreover, inferred peak global mean ocean temperatures during the LIG were ~1 to 1.5 ºC above preindustrial values (Shackleton et al., 2020). The LIG period can constrain the sensitivity of glacial systems to past natural warm periods, it will directly improve our ability to forecast future projections considering various climate scenarios. The LIG had a higher orbital obliquity (tilt angle of Earth's axis) relative to the current interglacial which leads to a positive annual insolation anomaly at high latitudes. During this period of warm climate global mean sea level (GMSL) was 6 to 9 m above present
(Dutton and Lambeck, 2012; Kopp et al., 2009a; Dutton et al., 2015). There are several relative sea level (RSL) reconstructions during the LIG which exhibit variable spatio-temporal structure, with some suggesting multiple sea-level highstands (Stirling et al., 1998; Hearty et al., 2007; Blanchon et al., 2009; Thompson et al., 2011; Dutton et al., 2015). Moreover, a relatively minor thermosteric sea-level contribution of less than 1 mESL tapers down into MIS 5e (McKay et al., 2011; Vaughan et al., 2013; Shackleton et al., 2020a). This suggests significant sea-level contributions from various sectors of the Greenland Ice Sheets and
AIS. Simulations of the Greenland Ice Sheet during the LIG have proposed a mass loss of 0.6 to 4.5 mESL (Tarasov and Peltier, 2003; Quiquet et al., 2013; Dahl-Jensen et al., 2013a; Helsen et al., 2013; Stone et al., 2013). The sea-level budget suggests an AIS contribution between 1.5 and 7.4 mESL during the LIG (Dutton et al., 2015), commonly attributed to the collapse of the WAIS.



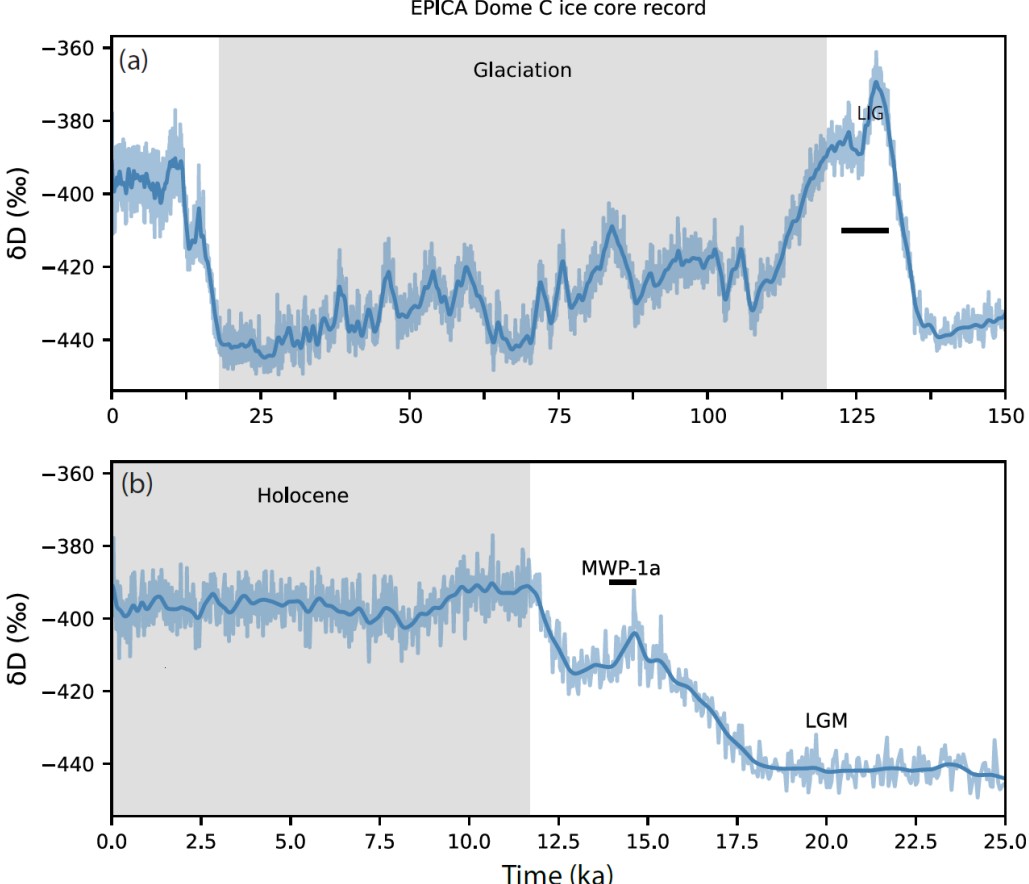

**Figure 2:** The EPICA Dome C deuterium record spanning (a) the time since the Last Interglacial (LIG), and b) the Last Glacial
Maximum (LGM), post-LGM deglaciation (including Meltwater Pulse 1a, MWP1a) and the Holocene.

Unfortunately, high quality constraints on the forcing and configuration of the AIS during the LIG are lacking. Additionally, previous modelling studies insufficiently explored parametric uncertainties and uncertainties in boundary conditions to robustly constrain the Antarctic contribution to the LIG sea-level highstand (Albrecht et al., 2020b; DeConto and Pollard, 2016a). There are little data constraining the chronology of AIS changes during the LIG. A recent study using octopus genome sequences suggested WAIS collapse during the LIG (Lau et al., 2023, doi: 10.1126/science.ade0664) but, so far, any direct evidence from proximal to the WAIS is ambiguous or under debate. Furthermore, the susceptibility of the various AIS sectors to change is effectively set by sub ice-shelf marine temperatures and circulation, both of which are very poorly represented in glaciological models especially in paleo contexts.

The LGM is the period of maximum global grounded ice volume, approximately 26 to 19 ka BP (Clark et al., 2009). However, the major continental ice sheets reached their respective local maximum grounded glacial volumes at different times, termed the local LGM (Clark et al., 2009). The local LGM of the AIS is poorly constrained and model reconstructions propose a range of values, while few AIS glacial simulations consider the available paleo observational data (Albrecht et al., 2020b; Briggs et al., 2013). Observational constraints on the past geometry of the AIS suggest a maximal but regionally variable LGM configuration around 20 ka (Livingstone et al., 2012; The RAISED Consortium, 2014). During the global LGM, GMSL was 120 - 134 metres below PD primarily due to the growth of large northern hemisphere ice sheets (Milne, Mitrovica and Schrag, 2002; Peltier and Fairbanks, 2006; Clark et al., 2012; Austermann et al., 2013; Lambeck et al., 2014). The first-order impacts on the spatial variability of sea-level change on millennial time-scales are due to GIA as a product of mass redistribution (e.g. grounded ice, ocean water, mantle flow) (Clark et al., 1978; Lambeck and Chappell, 2001; Milne and Mitrovica, 2008).

An outstanding issue regarding the LGM revolves around the question of missing ice to account for the GMSL low stand (Lambeck et al., 2014a; Clark and Tarasov, 2014; Simms et al., 2019). Studies reconstructing LGM ice sheet volumes during the LGM demonstrate a large variance. Near-field geological and geomorphological constraints on past ice sheet geometry apparently conflict



with the far-field RSL, as the former tend to favour smaller ice sheet volumes (Lambeck et al., 2014a; Clark and Tarasov, 2014; Simms et al., 2019). This could reflect potential issues in the interpretation of the living depth ranges of ancient corals since they might not be analogous to their present-day counter parts (Hibbert et al., 2016). Additionally, there remain uncertainties in

dynamic topography and GIA corrections (Austermann et al., 2013; Pan et al., 2022). More recently, in-situ radiocarbon ages from nunataks around the Ronne-Filchner ice-shelves have rejected a scenario that the LGM ice surface to the East of the Weddell Sea Embayment remained the same as present (Hillenbrand et al., 2014) but rather had thickened at the LGM by several hundreds of meters (Nichols et al., 2019), more consistent with an alternative LGM scenario of widespread grounded ice advance across the Weddell Sea shelf (Hillenbrand et al., 2014). The latest data on LGM ice surface height in the Weddell Sea sector could

constrain numerical simulations and enable larger AIS LGM volume than previously thought. By performing a large-ensemble history matching of the AIS since the LIG, inferential bounds for the LGM volume of the AIS will quantify the viability of larger Antarctic ice volumes and potentially diminish the sea-level budget shortfall or emphasize outstanding issues in the interpretation of the far-field RSL records.

GMSL rose throughout the post-LGM deglaciation with several distinct and abrupt accelerations in sea-level rise termed melt

water pulses (MWPs). The most pronounced event is MWP-1a at ~14.6 ka (Bard et al., 1990). The far-field RSL records exhibit a 15.7 to 20.2 m sea-level change over 500 years for MWP-1a (Deschamps et al., 2012; Carlson and Clark, 2012; Lambeck et al., 2014b; Lin et al., 2021). The Tahiti RSL record best constrains the magnitude and timing of MWP-1a and specifically suggests that it lasted for 300 years (14.6 to 14.3 ka) (Hanebuth et al., 2009; Deschamps et al., 2012). Models have often estimated MWP-1a sea-level contributions over a 500-year period rather than the shorter 300-year interval inferred by the Tahiti RSL record (Des-

champs et al., 2012). This implies that simulated MWP-1a sea-level contributions from individual ice sheets are likely overestimated. Historically, the MWP-1a budget shortfall had been attributed to an Antarctic contribution since it remains the least constrained of all the continental ice sheet volumes (Clark et al., 1996; Heroy and Anderson, 2007; Conway et al., 2007; Carlson and Clark, 2012). This was originally supported by geophysical GIA inversions of far-field RSL data which identified a significant Antarctic MWP-1a contribution (Bassett et al., 2005; Clark et al., 2002). More complete subsequent sea-level fingerprinting anal-

yses propose only a marginal contribution from the AIS to MWP-1a (Lin et al., 2021; Liu et al., 2016), which seems more consistent with the observational record (The RAISED Consortium, 2014). A few AIS modelling studies that were constrained by near-field observations found that the AIS had contributed a relatively small volume to MWP-1a (Albrecht et al., 2020b), although these studies performed a limited exploration of parametric uncertainties using 4 ensemble parameters. Through a large-ensemble history-matching methodology, we aim to quantify the AIS contribution to MWP-1a given near-field observational constraints to

better interpret past abrupt sea-level change.

In this study, an approximate history matching of the glacial system model (GSM) is performed against the updated observational AntICE2 database. We present model output on the evolution of the AIS since the LIG. Bounds on the AIS contributions to the LIG, LGM, and MWP-1a sea level are determined. Moreover, bounds on the AIS geometry through time are presented.

## 2. Observational constraints – AntICEdat 2.0

The Antarctic ICe sheet Evolution observational constraint (AntICE) database version 2 (henceforth referred to as the AntICE2 database) is used to evaluate Antarctic model reconstructions. The AntICE2 database is the most extensive collection of Antarctic paleo-data available (Fig. S1). It was recently expanded, recalibrated, curated, and discussed in detail in Lecavalier et al. (2023). The updated database partially built on the work of Briggs and Tarasov (2013). AntICE2 contains PD and paleo ice sheet constraints. The PD ice sheet configuration is constrained by BedMachine version 2 (Morlighem et al., 2020a) and surface velocity

(Mouginot et al., 2019). Additionally, there are PD observations which constrain contemporary and past AIS changes. These are ice core borehole temperature profiles and GPS uplift rate measurements. The remaining data consists of paleo-proxy observations of past AIS extent and thickness, and relative sea-level change. Excluding the PD state of the ice sheet, the AntICE2 database consists of 1023 high-quality observational data points that constrain past AIS evolution (Lecavalier et al., 2023). Fig. 1 and S1 illustrates the spatial distribution of the various data types and data identifiers. The available observational data enable

the identification of a sub-ensemble of simulations that are not-ruled-out-yet (NROY) by the data (often ambiguously referred to as best-fitting simulations in other studies).

The GSM history-matching analysis against the AntICE2 database is divided in two parts. Even though this study employs a joint/coupled ice sheet and GIA model, only data-model comparisons pertaining predominantly to ice sheet evolution are shown (paleoExt, paleoH, ice core borehole temperature, present-day geometry and velocity). Data-model comparisons to the GPS and

RSL data are relegated to part 2, where the results of the Antarctic GIA model are presented in detail (Lecavalier et al., 2024).

## 3. Model description

The Glacial Systems Model (GSM) has progressively undergone significant development to be suited for efficient millennial-scale AIS simulations. In this Section we present a short summary of the GSM and its various systems and components. The model



descriptions, developments, verification and validation experiments are discussed in greater detail in Tarasov et al. (in prep). The more recent model developments incorporated in the calibration include: 1) hybrid ice physics; 2) subgrid grounding line parameterization; 3) basal drag scheme; 4) ice shelf hydro-fracturing and ice cliff failure; 4) ocean temperature dependent sub ice-shelf melt parameterization; 5) subgrid ice shelf pinning point scheme; 6) expanded climate forcing scenarios; 7) expanded Earth rheology models for GIA. A diagram summarizing the major components of the Glacial Systems Model is shown in Fig. S2.

The ice dynamics in the GSM is based on the dynamical core of the Penn State University ice sheet model (PSU-ISM; Pollard and DeConto, 2007, 2009a). The PSU-ISM dynamical core was extracted, rendered modular, and coupled into the GSM. It consists of hybrid ice physics representing shallow ice and shallow shelf/stream regimes (SIA-SSA). The non-linear viscous flow of the ice is represented by Glen's flow law with a temperature-dependent Arrhenius coefficient (Cuffey and Paterson, 2013). To capture transient or steady-state grounding line (GL) migration involves resolving the GL (<200m resolution) or employing an analytical constraint on ice flux through the GL (Pattyn et al., 2012a; Drouet et al., 2012). The GSM employs a subgrid GL flux parameteri-

zation based on boundary layer theory (Schoof, 2007a). The parameterization relates the GL ice flux to longitudinal stress, sliding coefficient, and ice thickness. The subgrid interpolated depth-averaged ice velocity is imposed in the shelf flow equations.

The GL flux parameterization is defined for power law basal (Schoof, 2007a) and Coulomb plastic rheologies (Tsai et al., 2015). The GSM is configured to work with either a power law or Coulomb plastic basal drag parameterization. The underlying uncertainties of the ice-bed interface are incorporated in the basal drag coefficient which depends on basal temperature, hydrology,

basal roughness, and subglacial substrate, i.e., whether the ice is resting atop hard bedrock or unconsolidated sediment. The power law exponent is determined based on the substrate type since these basal environmental conditions yield different basal deformation. Alternatively, the basal drag over subglacial till can be represented using Coulomb plastic deformation (Tsai et al., 2015). The GSM basal drag component is broadly based on Pollard et al. (2015) and effective basal roughness derived from the basal topography subgrid standard deviation.

The basal drag coefficients can drastically impact ice sheet dynamics since they characterize ice deformation across the uncertain and poorly accessible basal environment. The GSM contains a dual basal drag scheme where ice deforming across a hard bedrock is described with a quartic power-law that jointly represents regelation and enhanced creep flow. To facilitate both basal deformation and rugosity of the soft till, basal drag schemes that characterize the various regimes are used (Schoof, 2005; Gagliardini et al., 2007; Tsai et al., 2015; Brondex et al., 2017, 2019). It has been shown that a power-law with sufficiently high basal drag

exponent can effectively represent a Coulomb-plastic scheme (Tulaczyk et al., 2000; Nowicki et al., 2013; Gillet-Chaulet et al., 2016; Joughin et al., 2019). Furthermore, Antarctic surface velocity assimilation studies concluded that till basal drag exponent exceeding five yields better agreement with observations (Joughin et al., 2019; Gillet-Chaulet et al., 2016). To represent all the compounding uncertainties affiliated with the till basal drag schemes, the till basal drag exponent in the GSM is chosen to be an ensemble parameter ranging between one and seven, which allows for a wide variety of till flow (Gillet-Chaulet et al., 2016; Nias

et al., 2018; Brondex et al., 2019; Joughin et al., 2019). Moreover, the GSM includes a Coulomb plastic till deformation-based derivation of the subgrid GL flux scheme (Tsai et al., 2015; Brondex et al., 2017, 2019).

The PD AIS loses a considerable amount of ice via iceberg calving (Depoorter et al., 2013;(Rignot et al., 2013). This is represented in the GSM using three calving components. The first component is based on crevasse propagation due to horizontal strain rate divergence and yields a calving rate (Winkelmann et al., 2011; Pollard and Deconto, 2012b; Pattyn, 2017; Levermann et al., 2012).

An additional parameterization contributes to the calving rate based on hydrofracturing, where surface meltwater or rain drains into crevasses. This additionally contributes to the strain rate divergence of the ice and helps propagate crevasses; thereby it increases the calving rate and can lead to potential ice shelf collapse (Nick et al., 2010). The third form of calving in the GSM is a tidewater ice cliff failure scheme (Pollard et al., 2015), this arises wherever exceedingly high ice cliffs experience an unbalanced horizontal stress gradient. Iceberg calving occurs when the overburden weight of the ice surpasses its yield strength, causing the

ice cliff to collapse (Bassis and Walker, 2012; Bassis and Jacobs, 2013; Pollard et al., 2015). The GSM applies a conservative approach to the ice cliff failure which prevents a cascading failure across an entire basin in only one model time step. This provides an allowance for the ice dynamics to adjust the geometry which can stabilize and buttress ice. The latter two calving components represent the Marine Ice Cliff Instability (MICI) where the hydrofracturing collapses an ice shelf which produces an unstable ice cliff (Pollard et al., 2015).

The most poorly constrained components of the glacial system are the surface climate and ocean forcing since the LIG. Most commonly, the climate forcing in ice sheet simulations is based on a single source, whether parameterized in the model or obtained from a single climate model (Golledge et al., 2014b; Albrecht et al., 2020a; Pittard et al., 2022). This neglects spatial variability and climate uncertainties which should be represented by an envelope of viable climate scenarios based on various climate reconstructions and inferences. In these instances, the resultant ice sheet simulations generate an envelope of outcomes which

are predominantly constrained by the chosen forcing. Therefore, three climate forcing schemes are implemented in the GSM to best represent an envelope of viable climate realizations. The three sets of climate fields are merged using ensemble parameter weights that blend the temperature and precipitation fields. The glacial index scheme uses a glacial index derived from the EPICA Deuterium record ($\delta D = \delta^2 H$) (EPICA, 2004; Jouzel et al., 2007). The first scheme is a parameterized climate forcing derived from PD monthly climatologies (RACMO 2.3p2; Melchior Van Wessem et al., 2018) and their relationship with lapse rate, elevation,

and glacial index, where an elevation threshold further controls precipitation. The second scheme uses PD monthly climatology





fields (Melchior Van Wessem et al., 2018) and Paleo-Modelling Intercomparison Project 3 (PMIP) glacial climatology fields (Bra-connot et al., 2012). The chosen LGM temperature and precipitation fields are the PMIP3 ensemble mean (excluding data-model misfit outlier) where temperature and precipitation empirical orthogonal functions (EOFs) are included to broaden the LGM degrees of freedom by capturing inter-model variance. The climate forcing is weighed back in time using the glacial index. The third scheme is based on a coupled energy balance model which applies top of atmosphere insolation with a glacial-interglacial ensemble parameter scaling constrained by Antarctic ice cores. The surface mass balance is then estimated using a positive degree day and positive temperature insolation surface melt scheme.

**Table 1:** Ensemble parameters in the Antarctic configuration of the Glacial Systems Model.

| Interface | Component | Parm # | Parm name | Definition |
|---|---|---|---|---|
| Ice dynamic | Basal env | 1 | rmu | Soft bed basal sliding coef. |
| Ice dynamic | Basal env | 2 | fslid | Hard bed basal sliding coef. |
| Ice dynamic | Ice deformation | 3 | fnflow | Glenn flow law enhancement factor |
| Ice - ocean | Calving | 4 | Ffcalvin | Calving coef. |
| Ice - atmosphere | Calving | 5 | pfactdwCrack | Geometric surface melt factor for hydrofracturing |
| Ice - ocean | Calving | 6 | CfaceMelt | Ice shelf face melt coef. |
| Ice dynamic | Basal env | 7 | wGF1 | Deep geothermal heat flux blending weight |
| Ice - atmosphere | Climate forcing | 8 | fnTdexp | Phase exponent of temperature |
| Ice - atmosphere | Climate forcing | 9 | fnpre | Glacial index scaling coef. for precipitation |
| Ice - ocean | SSM | 10 | fSSMdeep | Sub-shelf melt parameter |
| Ice - atmosphere | Climate forcing | 11 | fhPRE | Exponent for precipitation dependence on surface temperature change |
| Ice - atmosphere | Climate forcing | 12 | fnPdexp | Phase exponent of precipitation |
| Ice - atmosphere | Climate forcing | 13 | fnTdfscale | LGM scaling coefficient for glacial index |
| Ice - atmosphere | Climate forcing | 14 | rlapselgm | LGM temperature lapse rate |
| Ice - atmosphere | Climate forcing | 15 | fTweightPMIP | Mean PMIP3 temperature blending weight |
| Ice - atmosphere | Climate forcing | 16 | fPRE-weightPMIP | Mean PMIP3 precipitation blending weight |
| Ice - atmosphere | Climate forcing | 17 | fPEOF1 | LGM precipitation EOF field |
| Ice - atmosphere | Climate forcing | 18 | fTEOF1 | LGM temperature EOF field |
| Ice - atmosphere | Climate forcing | 19 | fTEOF2 | LGM temperature EOF field |
| Ice - atmosphere | Climate forcing | 20 | fnTEBMscale | Energy balance model scaling |
| Ice - atmosphere | Climate forcing | 21 | fTweightEBM | Energy balance model temperature blending weight |
| Ice dynamic | Basal env | 22 | Fbedpow | Till fraction exponent for bed classification and basal drag adjustment due to fractional till |
| Ice - ocean | SSM | 23 | TregSSMCut0 | Default ocean temperature bias corr. |
| Ice - ocean | SSM | 24 | TregSSMCut1 | Ross sector ocean temperature bias corr. |



| Ice - ocean | SSM | 25 | TregSSMCut2 | Amundsen sector ocean temperature bias corr. |
|---|---|---|---|---|
| Ice - ocean | SSM | 26 | TregSSMCut3 | Ronne sector ocean temperature bias corr. |
| Ice - ocean | SSM | 27 | TregSSMCut4 | Filchner sector ocean temperature bias corr. |
| Ice - ocean | SSM | 28 | TregSSMCut5 | Amery sector ocean temperature bias corr. |
| Ice dynamic | Basal env | 29 | POWbtill | Soft bed power law exponent |
| Ice dynamic | Basal env | 30 | fSTDtill | Sub-grid roughness dependency parameter for soft bed sliding |
| Ice dynamic | Basal env | 31 | fSTDslid | Sub-grid roughness dependency parameter for hard bed sliding |
| Ice - ocean | SSM | 32 | rToceanPhase | Glacial index exponential phase factor for Tocean |
| Ice - ocean | SSM | 33 | rToceanWrm | Scaling factor for negative glacial index |
| Ice dynamic | Basal env | 34 | wtBedTill1 | Basal till fraction blending weight |
| Ice dynamic | Basal env | 35 | rHhp0 | Grounding line parametrization selection |
| Ice - solid Earth | GIA | 36 | earthLT | Lithosphere thickness |
| Ice - solid Earth | GIA | 37 | earthUV | Upper mantle viscosity |
| Ice - solid Earth | GIA | 38 | earthLV | Lower mantle viscosity |

The other dominant method by which the PD AIS undergoes negative mass balance is through sub-ice-shelf melt (SSM) (Rignot et al., 2013; Depoorter et al., 2013; Liu et al., 2015). The GSM calculates sub-ice-shelf mass balance via an ocean temperature dependent parameterization at the ice-ocean interface (Tarasov et al., In prep). This calculates mass balance at the ice front,
beneath the ice shelves, and at the grounding line. The ocean temperature forcing is based on transient TRACE-21ka simulations (He, 2011) which are PD bias corrected by the Estimating the Circulation and Climate of the Ocean (ECCO) reanalysis ocean temperatures (Fukumori et al., 2018). There are six key major marine-terminating sectors from which ocean temperature profiles are extracted: Ross Sea sector, Amundsen Sea sector, Antarctic Peninsula, Weddell Sea sector, Amery ice shelf basin, and East Antarctic coast (Dronning Maud Land, Wilkes Land - Victoria Land coasts). The ocean temperature profiles are extrapolated be-
neath the ice shelves with a cut off defined by the minimum sill height when dealing with deeper continental shelves. As the changes in sub-ice shelf ocean temperature during the LIG have a critical impact on the resulting LIG sea-level high-stand and to avoid extrapolating TRACE ocean temperatures for warmer conditions, a separate ensemble parameter is introduced. Given the relationship between Antarctic $\delta^2$H and mean ocean temperature (Shackleton et al., 2021), this parameter (rToceanWrm) simply scales the glacial index derived atmospheric warming and adds it to the PD ocean temperature climatology. The deep-sea benthic
foraminifera stack represents a proxy for deep ocean temperatures and global grounded ice sheet volume during the past (Lisiecki and Raymo, 2005). Within the GSM, the benthic stack and RSL observations drive the far-field global sea-level forcing (Lambeck et al., 2014b).

One of the primary initialization conditions is the PD AIS geometry - basal topography, ice thickness, and ice surface elevation. The Antarctic GSM configuration uses the Antarctic BedMachine version 2 (Morlighem et al., 2020b). The poorly-observed basal
environment remains a major source of uncertainty to ice sheet evolution. There are several key basal boundary conditions: the basal topography, geothermal heat flux, and subglacial substrate type (i.e. sediment distribution). The ice sheet is externally forced at its base by the geothermal heat flux. There are sparse measurements and inferences made at ice core sites that reached the bed (Pattyn, 2010). To partially account for uncertainties in the geothermal heat flux, an envelope of realizations is produced by blending two fields using an ensemble parameter. The first geothermal heat flux field is based on the spectral analysis of
airborne magnetic data (Martos et al., 2017), while the other complementary field is based on the thermoelastic properties of seismic data in the crust and upper mantle (An et al., 2015).

With respect to the substrate type distribution beneath the AIS, an elevation-based approach is used to infer the till fraction which effectively controls the basal drag. An elevation-based approach generally postulates that unconsolidated material, i.e. subglacial till and/or fossil marine sediments, prevails in areas below sea level, whereas hard bedrock dominates in areas above
sea level (Studinger et al., 2001; Pollard and DeConto, 2009b; Martin et al., 2011). The most probable regions with infill of marine





sediments are those below sea level prior to large scale glaciation across Antarctic with a glacial isostatic equilibrated topography (e.g., Studinger et al., 2001). Over the course of many glacial cycles, the ice sheet transported detritus eroded from elevated bedrock down to submarine sectors. However, at present there are many features beneath the ice sheet that have survived successive glaciations, thus some features below sea level are presumed to be composed of hard bedrock, too (e.g., Bingham et al., 2017; Alley et al., 2021). The GSM is geared to avoid potential overfitting issues to the PD geometry since our aim is to confidently bracket past and present transient changes. Hence, we avoid a basal drag inversion scheme to infer basal drag coefficients since many processes are integrated in these coefficients. Therefore, to maximize long term retrodictive capabilities, the GSM uses a fully unloaded glacial isostatic equilibrium sea-level threshold scheme. Additional considerations must be made to account for dynamic topography (Austermann et al., 2015, 2017). Uncertainties in dynamic topography on a 35 Myr timescale can significantly impact the range of viable sea-level elevation thresholds for determining probable subglacial sediment distributions. Regional elevation thresholds ranging between -300 to -100 m are justified given the spatial variability in dynamic topography and its uncertainties. The regional thresholds are selected based on first principles where deep subglacial basins/troughs, and regions of fast-flowing ice exceeding 400m/yr are properly delineated as being underlain by soft till. To properly classify crucial pinning points and local maxima in basal topography as highly consolidated sediment and hard bedrock, respectively, the thresholds are refined to properly delineate key pinning features. After the first few waves of ensembles results, persistent outstanding PD ice thickness misfits were related to the misattribution of the subglacial substrate type distribution. These persistent PD misfits were used to perform an update to the substrate distribution.

Pinning points that often manifest as ice rises and ice rumples can significantly affect GL dynamics (Favier et al., 2012, 2016; Berger et al., 2016; Wild et al., 2022). Ice shelves are buttressed by various topographical features; however, many crucial pinning points are inadequately resolved in model simulations due to resolution limitations. This is particularly relevant because small ice shelf pinning points can significantly influence transient ice dynamics and grounding line migration (Favier et al., 2012, 2016). The GSM uses a subgrid statistical pinning point parametrization scheme to rectify these limitations. Unresolved subgrid features must be represented since they produce characteristic features at the PD AIS surface, such as ice ridges, rumples, and rises, that buttress the ice by generating substantial basal stresses that impact upstream flow. Since subgrid pinning points have been preserved through many consecutive glaciations, they must consist of hard bedrock. Therefore, to enhance the subgrid pinning points and prescribe their hard bed geomorphology, the till sediment fraction is exponentiated. Originally, the till fraction is upscaled from the Antarctic BedMachine native resolution of 500x500m to 40x40km. The upscaling emphasizes or de-emphasizes certain subgrid pinning point features depending on their scale, geometry, and how they are distributed against the model grid. The subgrid pinning point enhancement exponent is varied regionally between 1 and 12 to enhance the till fraction value of subgrid features that are currently pinning ice across the present-day ice sheet (Tarasov et al., in prep).

The GSM is coupled to a glacial isostatic adjustment model of sea-level change. The GIA component is based on a spherically-symmetric visco-elastic gravitationally self-consistent Earth model which calculates GIA due to the redistribution of surface ice and ocean loads (Tarasov and Peltier, 2004). The Earth model rheology has a density structure based on preliminary reference Earth model (PREM) (Dziewonski and Anderson, 1981) and an ensemble parameter controlled three-shell viscosity structure. The viscosity structure is defined by the depth of the lithosphere, upper and lower mantle viscosity. The GIA component shares many similarities to that used in Whitehouse et al., (2012c) for post-process glaciological model runs. However, our GIA component is fully coupled to the ice sheet model and includes broader parametric uncertainties. The GIA calculations are computed every 100 simulation years. To minimize the considerable computational cost of solving for a complete gravitationally self-consistency solution coupled with an ice sheet model (Gomez et al., 2013), a linear geoidal approximation is used to account for the gravitational deflection of the sea surface. However, upon completing the full transient simulation, a gravitationally self-consistent solution is computed for determining RSL and vertical land motion.

The Antarctic GSM domain is polar stereographic with a horizontal model resolution of 40 by 40 km. The vertical model resolution has 10 layers unevenly spaced when dealing with ice dynamics, while the thermodynamic component uses 65 vertical layers. The Antarctic configuration of the GSM consists of 38 ensemble parameters which is the most comprehensive representation of uncertainties in the Antarctic glacial system of any study to date. A given simulation is defined by the chosen values of the ensemble parameters, referred to as a parameter vector. Model parameters which exhibited no significant sensitivity on the model outcome for a diverse set of reference parameter vectors were dropped from being included as ensemble parameters. The ensemble parameters define the uncertainties in the climate forcing, mass balance, ice dynamics, and GIA (Table 1). The ensemble parameter history-matched distributions are shown in Figure S3 to S7.

# 4. Methodology

## 4.1 Scoring a reconstruction

For a given full transient simulation, the resulting AIS reconstruction is compared to the present-day ice sheet geometry on the simulated grid and several scores are produced. Using the Antarctic BedMachine version 2 dataset (Morlighem et al., 2020b), a thickness root-mean-square-error (RMSE) for the WAIS (which includes the Antarctic Peninsula Ice Sheet for simplicity), the East



Antarctic Ice Sheet (EAIS), and floating ice are separately calculated considering uncertainties in the BedMachine inferences. Moreover, an RMSE is calculated for the PD ice shelf area and PD GL position score along 5 transects (shown in Fig. 1). Using the MeaSUREs PD surface velocity dataset (Mouginot et al., 2019), a RMSE is calculated for surface velocities in the interior and margin of the ice sheet as defined by a 2500 m surface elevation threshold. The ice sheet simulation is then scored against the data described in AntICE2, with a predominant focus on tier-1 and 2 data. Tier-1 data is the highest quality data which has the

greatest power to constrain the ice sheet and GIA model. While tier-2 data supplements tier-1 by providing more granular detail on past changes. Tier-3 data is excluded from the history-matching analysis since it correlates highly with the higher quality tier-1/2 data (Lecavalier et al., 2023).

The ice core borehole temperature profiles are scored by extracting a PD temperature profile from the reconstruction at each borehole site. A given borehole temperature can be broadly described by five observations: 1) depth of profile; 2) ice thickness

at the borehole site; 3) near surface temperature, 4) englacial temperature, and 5) basal borehole temperature. Typically, there are ice thickness mismatches with the observed PD ice thickness, therefore, the simulated borehole temperature profile depth must be rescaled to match the observed borehole depths. The englacial temperature comparison was performed at the englacial temperature minima which aligned most closely with the GSM vertical grid ice temperature output. Subsequently, the RMSE from the near surface, englacial, and basal temperature is calculated to infer a score for a given borehole temperature profile. A

quadrature is calculated of all the borehole temperature profiles to obtain a borehole temperature profile score for a given simulation.

Using the Antarctic BedMachine basal topography and the AntICE2 cosmogenic exposure ages, the paleo ice thickness (paleoH) data can be directly compared to an AIS simulation. The model produces a chronology for ice thickness changes across the entire Antarctic continent, and changes in ice thickness are extracted at each respective paleoH data site. For a given paleoH observa-

tion, the nearest simulated ice thickness value is identified in space and time. Considering structural uncertainties, the neighbouring spatial grid cells (± 40 km) and time steps (± 500 yrs) are accounted for in the paleoH scoring error model. The quadrature of all residuals based on the simulated and observed past ice thickness given uncertainties is calculated to generate a paleoH score. The past ice extent (paleoExt) score is similarly calculated as in paleoH score, except it considers the timing that a grid cell is covered by ice, when that ice becomes ungrounded, and when the grid cell is deglaciated. This enables a broader comparison

to the paleoExt database which includes proxy data for proximal to GL (PGL), sub-ice-shelf (SIS), and open marine conditions (OMC).

When a joint AIS and GIA simulation is completed, a full gravitationally self-consistent GIA simulation of sea-level change is performed over the last glacial cycle. This provides RSL and PD bedrock deformation rates which can be compared to the AntICE2 paleoRSL and GPS database. These results have consequences on AIS evolution and are integrated in the results presented in this

study. Comprehensive data-model scoring details can be found in Tarasov et al. (in prep).

## 4.2 History-matching analysis

This study involves a history-matching analysis of a complex system against observational constraints of various datatypes to rule out simulations which are inconsistent with the data (Tarasov and Goldstein, 2021). History matching requires a full accounting of uncertainties, though the error models for quantifying these uncertainties can be specified much more freely than required

for a full Bayesian Inference. A history-matching analysis and initial model calibration consist of ruling out model reconstructions which are unequivocally inconsistent with the observational constraints to produce a state-space estimation of the AIS which brackets the true ice sheet history. This yields a sub-ensemble of model simulations that are not inconsistent with the data within a threshold, and thereby provide approximate bounds on the probable evolution of the AIS since the LIG.

**Table 2:** The full ensemble and subensemble descriptions.

| Ensemble name | # of members | Description |
|---|---|---|
| ANtot | 27,500 | All previous AIS waves of ensembles leading up to final waves |
| AN | 9,293 | Full ensemble – final wave of ensembles |
| AN4sig | 973 | Sub-ensemble of AN sieved to be 4σ of AntICE2 |
| AN3sig | 82 | Sub-ensemble of AN sieved to be 3σ of AntICE2, except 3.5σ of paleoExt data and floating ice RMSE, and 4σ of paleoRSL data |

As part of this study, several large-ensemble data-constrained analyses were iteratively performed to evaluate the model's ability to bracket AntICE2. GSM simulation output was applied towards supervised machine learning of Bayesian Artificial Networks (BANNs) for Markov Chain Monte Carlo (MCMC) sampling to efficiently explore relevant portions of the parameter space.



A flow chart is shown in Figure S8 that illustrates the history-matching algorithm. The appearance of significant data-model discrepancies that persist after converged history-matching waves is generally indicative of insufficient sampling of the model parameter space and/or underestimated uncertainties in data and/or model. Given our sampling approach and care in constraint database specification, this problem was indicative of insufficiently specified model structural uncertainty. When structural uncertainties were so large that they were deemed unacceptable, the model degrees of freedom were expanded, and refinements

were made to model components and inputs. This included revising the subglacial substrate type distribution, pinning point, and basal drag schemes, as well as broadening the geothermal heat flux ranges and defining distinct marine basins to parametrize regional ocean forcing. This necessitated a series of repeated history-matching cycles, culminating in ~40,000 AIS simulations over the last two glacial cycles. The methodological details of this work are specified in Tarasov et al. (in prep). In this study we present the most relevant final waves of ensembles which consist of 9,293 simulations. We will refer to these as the "full ensem-

ble" (Table 2).
  Our initial understanding of the glacial system is encapsulated within the ensemble parameter prior probability distribution ranges. The distributions are based on previous studies, expert judgement, and are initially kept wide as not to miss any potentially viable ensemble parameter combinations. The data-model comparison is characterized by the error model which combines all the errors from the observational and structural uncertainties. Observational data include data-system uncertainties that are

composed of measured uncertainties and uncertainties affiliated with the indicative meaning of the proxy. Structural uncertainties are irreducible (in that they cannot be reduced by a more appropriate choice of model ensemble parameters) and are nontrivial to specify because they represent the model deficiencies with respect to reality. The structural uncertainty must be carefully defined and not underestimated since underestimated uncertainty will invalidate inferential bounds (Tarasov and Goldstein, 2021).

Internal discrepancy is the component of structural uncertainty that can be quantified by numerical experiments. The internal discrepancy analysis conducted on the GSM involved assessing the impact of uncertainties from basal topography, geothermal heat flux, climate forcing, sea-level forcing, and initialization. This is evaluated by experimenting on a high variance set of reference parameter vectors using a wide variety of boundary conditions with noise super imposed to bound their respective uncertainties. This defines a variance or covariance matrix of the internal discrepancy multivariate distribution. The internal discrep-

ancy analysis yields an uncertainty contribution and bias contribution for each of the data type scores. The external discrepancy of the model cannot be inferred directly through model experimentation and is particularly challenging to define. The main structural uncertainties affiliated with the GSM are model approximations (e.g. hybrid physics, parameterizations), grid resolution and subgrid processes. As an initial estimate, the external discrepancy bias and uncertainties are assigned a large value so as to not underestimate structural error. The value is consequently refined/narrowed over successive ensemble iterations. The

structural error assessment will be described in detail in a future publication.
  The observational error model has a Gaussian distribution which assumes minimal spatiotemporal error correlation between observations. The AntICE2 observational dataset was curated for quality over quantity with the objective of also minimizing the multivariate structure of the error correlation. Some of the more significant error correlation is associated with the age calibration and corrections of the data (e.g. $C^{14}$ and $Be^{10}$ dating, reservoir corrections). Moreover, the data-model comparison needs to

account for the uncertainties affiliated with transposing the exact location of the data, i.e., the geographical location of a sample, onto the coarse model grid, such that a meaningful comparison can be made. This involves evaluating model output from neighbouring grid cells of the data's transposed location to ascertain whether any deficiency is a result of structural errors associated with resolution dependencies.
  We address the fact that parameter space cannot be exhaustively evaluated (because it is computationally intractable) by per-

forming a Markov Chain Monte Carlo (MCMC) sampling of the parameter space to evaluate the most relevant portions of the parameter space which performs well against the AntICE2 database. Hundreds of nominally converged MCMC chains initiated from dispersed regions of the prior distribution were performed since the GSM is a non-linear system with high dimensionality, and a single chain could potentially only evaluate a local high scoring region in the parameter phase-space.

## 4.3 Ensembles and not-ruled-out-yet (NROY) sub-ensembles

Several large ensembles of simulations were conducted (~40,000 members), and their output was compared against observational constraints. The full ensemble of simulations is iteratively expanded through successive waves of new simulation ensembles. The latest ensemble waves are used to progressively rule out unlikely parameter vectors that significantly misfit the observations beyond chosen multiples of the total uncertainty (internal discrepancy, external discrepancy, and data uncertainty). This involves defining thresholds for each implausibility component. In the case of the Antarctic GSM configuration, the metrics of

interest were chosen to be: present-day ice thickness root-mean-square-error for WAIS, EAIS, and ice shelves; present-day ice shelf area score; present-day GL position score along 5 transects; ice core borehole temperature profile score; GPS uplift rate score; past ice thickness score; past ice extent score; and past relative sea level score. The datatype implausibility thresholds are based on the Pukelsheim 3σ rule which states that 89% of the probability density for any continuous distribution is within 3σ of the mean. Directly applying a 3σ cut-off yielded just a few plausible runs, therefore, a broadening to a 4σ of the total uncertainty

threshold for all datatype scores was applied (AN4sig: N=973). A 3σ of the total uncertainty threshold was then applied on all



datatype scores when sieving for best-fitting sub-ensembles but allowing past ice extent (3.5σ), ice shelf RMSE (3.5σ), and relative sea-level scores (4σ) a less restricted threshold (AN3sig: N=82). This larger allowance with these three scores was justified given the model struggles to bracket a few observations in these data types, which previously resulted in ruling out nearly all simulations. These thresholds are then used in the history-matching sieve to impose an implausibility measure to rule out simulations
(Table S1). The sub-ensemble not ruled out consists of simulations and parameter vectors which define the basis for BANN training and GSM emulation. MCMC sampling of the BANNs is used to propose new parameter vectors that make up subsequent waves of ensembles for history matching. Each successive wave of ensembles refines the regions of the parameter space that reasonably fit the observations. These ensembles are further used to revise the emulators for MCMC sampling. The iterative process of incorporating additional ensembles and subsequent history matching, defines and expands the NROY ensemble pa-
rameter space.

Initially the prior distributions for the ensemble parameters were chosen to be uniform, or quadratic functions favouring the top, bottom, or middle values of the parameter range. Wide prior distributions were determined with ranges physically motivated or taken from the literature. Secondary narrow prior distributions were defined to sample regions which are more commonly assigned in the literature. A dispersed random sampling of the ensemble parameters based on Latin Hybercube sampling was
initially conducted using both wide and narrow prior distributions. The majority of these initial simulations performed quite poorly, with a limited few approaching the PD geometry. From these initial ensembles, few selected runs were chosen as initial reference simulations and parameter vectors. A sensitivity analysis was performed across the GSM ensemble parameters using this set of reference parameter vectors to evaluate the relative impact of various ensemble parameters.

The ensemble thus far was then sieved to isolate the best ~10% of simulations. The initial best fitting sub-ensemble was then
used to fit beta distribution parameters for each ensemble parameter. From these beta distributions a series of parameter vectors was generated that ideally produced better performing AIS reconstructions. The full ensemble was carefully evaluated against the AntICE2 database and PD observations to verify that the observations are adequately bracketed within uncertainty. This initially led to a revision of ensemble parameters, model developments, and revisions to certain boundary conditions. Considering all the simulations leading up to the final waves of ensembles, all previous experimentation, sensitivity analyses, Latin
Hypercube and beta fit sampling consisted of ~30,000 model simulations (total ensemble ANtot minus full ensemble AN). Unfortunately, when a model undergoes significant model development, much of the previous model results lose relevance because they are based on a different model configuration which exhibits different behaviour than the latest version. Beyond those efforts, additional Latin Hypercube and beta distribution sampling was carried out before training BANNs and MCMC sampling. In Section 6, we present the latest waves of ensemble results based on the history-matching large-ensemble data-constrained analysis (full
ensemble AN with N=9,293; see Table 2).

## 5. NROY fits to data constraints

In this study we conducted ~40,000 AIS reconstructions since the LIG and present the results from the final ensembles consisting of 9,293 reconstructions. Using the observational constraint database AntICE2 and a history-matching methodology, the full ensemble is reduced to a sub-ensemble that is NROY by the data. The full ensemble is sieved such that runs must perform beyond
a specified 4σ/3σ threshold across all data type scores. The full ensemble is reduced to a sub-ensemble representing the best-fitting reconstructions when compared to the AntICE2 observational constraint database (termed the NROY AN3sig sub-ensemble consisting of 82 reconstructions). The NROY bounds presented in this study are those defined by the entire AntICE2 database, alternate bounds can be produced which target a subset of the AntICE2 database to explicitly focus on specific research objectives (e.g. targeting PD observations or jointly targeting paleoRSL and GPS data).

Here we present the data-model comparison of the full ensemble, NROY AN3sig sub-ensemble, and a high variance subset (HVSS) selection from AN3sig sub-ensemble, with the latter being integrated within the GLAC3 global ice sheet chronology for future analysis. A HVSS of 18 simulations was extracted from the NROY AN3sig sub-ensemble to showcase some glaciologically self-consistent simulation results. The HVSS simulations are shown against the LIG and LGM metrics of interests in Figure S9. Three simulations are showcased from a HVSS from the NROY sub-ensemble, they collectively represent the best-fitting simulation with
varied LGM and LIG grounded ice volume anomalies. (RefSim1, RefSim2, RefSim3 being the reference simulation with run identification number nn61639, nn60138, nn61896, respectively). Summary of key data-model comparisons are shown in Fig. 3-8, while the remaining comparisons are found in Lecavalier et al., (2024).





**Figure 3:** Ice core borehole temperature profile data-model comparison where the grey shading are the full ensemble statistics. The solid and dashed black lines are the mean and min/max ranges for the not-ruled-out-yet (NROY) best fitting AN3sig sub-ensemble. Simulations consisting of a high variance subset (HVSS) of the NROY AN3sig sub-ensemble are shown in red. Site a-h) and n-p) are high quality tier-1 temperature profiles; i-k) are tier-2 profiles since they correlate significantly with the Siple Dome profile; and l-m) are lower quality tier-3 profiles which only partially span the ice column. The 2σ and 1σ ranges are the nominal 95% and 68% ensemble intervals based on the equivalent Gaussian quantiles, respectively.

## 5.1 Ice core borehole temperature profiles

Many processes impact the temperature of Antarctic ice through time. Even though the temperature profiles were acquired in the late 20th and early 21st century, the temperature profiles contain a substantial amount of integrated information about past ice sheet changes, atmospheric forcings, the geothermal heat flux, and basal conditions, since temperatures propagate through the ice slowly (Cuffey and Paterson, 2010). Generally, the borehole temperature profiles can be categorized into two groups, 1) those whose near surface temperatures are clearly the coldest across the entire profile (e.g. EPICA Dome C), and 2) those whose englacial temperature remain as cold as near surface ice temperatures (e.g. WAIS Divide); generally these two categories reflect low and high rates of snow accumulation, respectively, and corresponding rates of downward advection of cold surface ice. Broadly speaking, the full ensemble brackets the ice core borehole temperature profiles with NROY sub-ensemble simulations effectively capturing the observed data (Fig. 3). The model reproduces both categories of temperature profiles. The ensemble results can explain these types of profiles by identifying the dominant forcings and processes which impact the temperature



profiles. Firstly, the geothermal heat flux warms from the base, a primary energy flux impacting basal ice temperatures and whether basal ice reaches the pressure melting point. Places with a warm bed tend to experience higher ice velocities, which draws in surrounding ice. Atmospheric temperatures and incoming radiation directly force the surface of the ice sheet where the firn layer buffers temperatures before conducting temperatures directly into the surface ice. Ice dynamics will advect ice which will perturb the temperature profile, this can displace colder ice from the surface deeper into the ice column. When evaluating the best-fitting NROY sub-ensemble, the temperatures of type 1 profiles tend to remain clustered relatively close to the observations. Conversely, the NROY sub-ensemble results at type 2 profiles show significant variance. Simulations that produce cold englacial temperatures, achieve this because cold surface and surrounding ice is advected at depth.

The simulated temperature profiles are scaled to the observed ice thickness at each borehole site to properly compare the simulation results to the observations. Notable outstanding misfits with respect to the full ensemble and NROY sub-ensemble remain. The interior of the EAIS has four high quality borehole temperature records (EDC, Vostok, Dome Fuji, and EPICA Dronning Maud Land; tier-1 sites Fig. 3a-d) and one lower quality partial borehole record at the South Pole (tier-3 site Fig. 3m). The NROY AN3sig sub-ensemble simulations capture the observations in the EAIS interior with a few exceptions. The AN3sig simulations in this region tend to favor warmer temperatures near the surface and cooler temperatures at depth with respect to the observations, suggesting issues with the implemented PD reanalysis climatology and/or PD elevation mismatches. The simulated temperatures near the bed narrowly capture the observed temperatures or are insufficiently warm, such as at Dome Fuji, where neither the full ensemble nor the NROY ensemble get warm enough at depth. These deficiencies are likely a product of the surface and basal thermal forcing. However, in previous ensemble waves attempts were made to address the cold basal ice issue with limited success. The geothermal heat flux is based on a magnetic (Martos et al., 2017) and seismic inferences (An et al., 2015), and a weight ranging between 0 and 1 is used to blend the fields. The degrees of freedom in the geothermal heat flux (GHF) boundary condition were expanded by allowing for a weight marginally greater than 1 to enable a broader range of GHF values. Albeit, the extrapolated GHF fields remained bounded by their inferred uncertainties to prevent entirely unphysical values. Ultimately, this partially addressed basal misfits but at some sites the proposed range of GHF values between the magnetic and seismic inferences were too similar to sample a sufficiently wide range of potentially viable GHF (e.g. Dome Fuji). This points to the need for more complete inferences of the GHF field especially on the uncertainty side. Especially troubling are the lack of uncertainty range overlap for key GHF inference for some regions.

The borehole temperature profiles in the WAIS interior are clearly type-2 profiles with cold englacial temperatures (WAIS Divide, Byrd; tier-1 sites Fig. 3e-f). The full ensemble AN and NROY AN3sig sub-ensemble are capable of producing cold temperatures at depth, however, with a large variance of simulation outcomes with limited simulations reproducing the observed profile. At the WAIS Divide borehole site, the simulations tend to favor warmer basal temperatures with respect to the observations and again highlight potential limitations in simply blending two GHF inferences with similar inferences at a given site. This results in a narrowed exploration of GHF values at certain sites. Several borehole temperature profiles have been obtained from ice streams along the Siple Coast and from Siple Dome. These profiles correlate with each other. The Siple Dome borehole profile is the local high-quality tier-1 representative for the region (Fig. 3g), while the temperature profiles from the ice streams are relegated to tier-2 status (Fig. 3i-k). The full ensemble and NROY sub-ensemble both capture the ice stream temperature profiles. The full ensemble manages to bracket the Siple Dome temperature profile, however the NROY sub-ensemble remains too warm at the surface and base. This is likely due to the misrepresentation of the local ice dome due to resolution limitations, with the model having better ability to resolve the ice streams on the Siple Coast. Thus, the modelled ice thickness in this region is generally less than the PD ice thickness, which in turn leads to warmer surface temperatures overall.

There are several other temperature profiles near the PD GL, Law Dome, Talos Dome, Fletcher Promontory, Skytrain Ice Rise, and Berkner Island (Fig. 3h,l,n-p). These are all high-quality temperature profiles (tier-1), with the exception of the partial temperature profile at Talos Dome (tier-3). The borehole sites are located near or around topographic and basal features which are poorly resolved in the GSM. The full ensemble brackets the observed profiles at the Law Dome, Talos Dome, Skytrain Ice Rise, and Berkner Island, albeit not by the 2σ range. The NROY sub-ensemble fails to bracket the observations at these borehole sites. Additionally, at the Fletcher Promontory, the simulated temperatures are far too warm at the base. Considering these borehole sites are surrounded by complex basal topography that is poorly resolved, the analysis prioritized to capture the temperatures profiles in the interior of the ice sheet.

The GHF BC inferences are spatial fields and a chosen weight parameter might improve the fit at one site but directly decrease the fit at another. Therefore, future work will focus on broadening the degrees of freedom in the GHF BCs to enable some additional spatial variability beyond the GHF inferences to explore a broader range of potentially viable GHF values across borehole sites that are too warm or too cold with respect to the observations. Additionally, due to mismatches in PD ice thickness between observed and simulated at the borehole sites, this directly leads to misfits in surface ice temperature which should be factored into the scoring calculations. Otherwise, one is double counting misfits across multiple data types (borehole temperatures and PD ice thickness).



## 5.2 Past ice extent


The full ensemble of simulations is compared to observations of past ice extent that are in the AntICE2 database. The data-model comparison is performed against tier-1 and 2 observations which includes proximal to the GL (PGL), sub-ice-shelf (SIS), open marine conditions (OMC) (ages shown in in Fig. S103 of Lecavalier et al., 2023). However, this discussion will focus on the data-model comparison with the highest quality data only (tier-1 data-model comparison; Fig. 4). Most past ice extent data are brack-

eted by the full ensemble and NROY sub-ensemble, with a few noted exceptions.

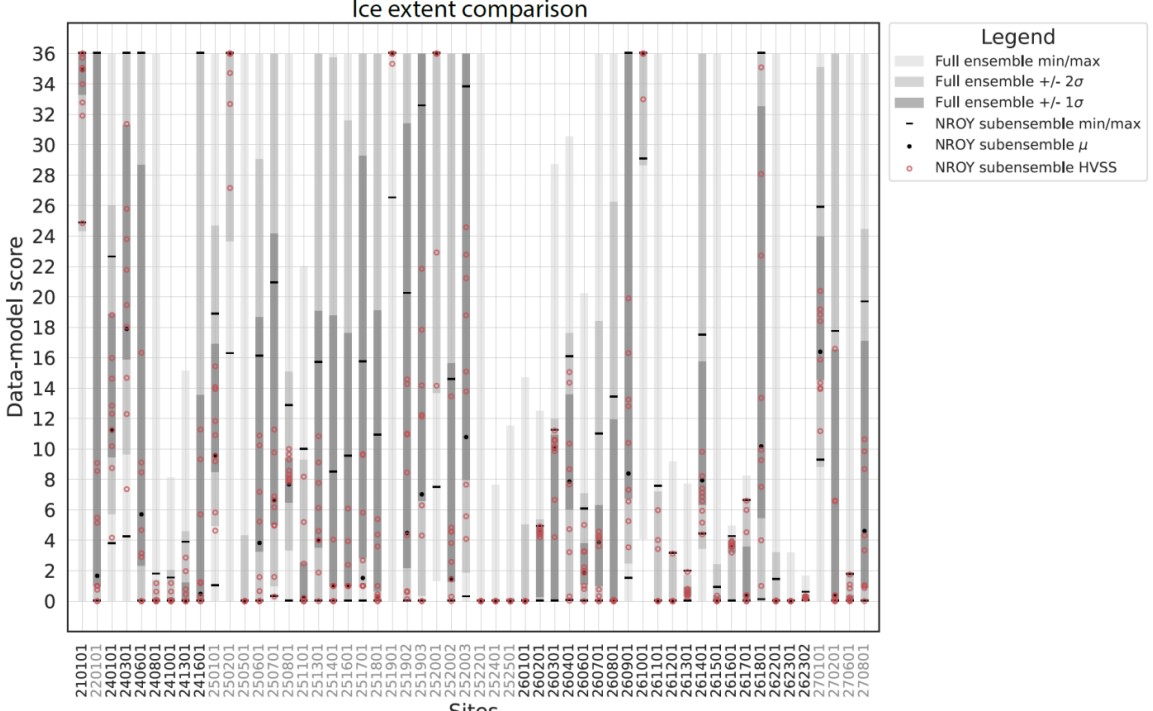

**Figure 4:** Past ice extent data-model comparison scores for the highest quality tier-1 data in AntICE2. The grey shading represents the min/max, 1σ and 2σ ranges of the full ensemble. The solid black circles and lines are the mean and min/max ranges for the not-ruled-out-yet (NROY) AN3sig sub-ensemble. Simulations consisting of a high variance subset (HVSS) of the NROY AN3sig sub-

ensemble are shown as red circles. The 2σ and 1σ ranges are the nominal 95% and 68% ensemble intervals based on the equivalent Gaussian quantiles, respectively. The AntICE2 paleoExt data ID are shown in Figure S1.

Additionally, the GSM simulations are compared to the reconstructions by The RAISED Consortium (2014), which was a large community effort with expert interpretations of a variety of data types. Even though there has been more observational data collected in the decade since the initial RAISED Consortium effort, the NROY AN3sig sub-ensemble ice extent statistics are com-

pared to the reconstructions published in The RAISED Consortium (2014) in Fig. 5. The RAISED Consortium (2014) binned their ice extent contours to the nearest 20, 15, 10, or 5 ka interval, which makes their speculated and inferred ice extent contours somewhat ambiguously defined from the raw observations.

East Antarctica has limited ice extent observations with only three constraints for all of Dronning Maud – Enderby Land, Lambert – Amery, and Wilkes – Victoria Land sectors combined. In the Dronning Maud – Enderby Land sector, OMC near the PD ice shelf

edge is dated at the turn of the Holocene (site 2101; 11.6 ka). The ice shelf in this area is buttressed by prominent pinning points which are poorly resolved by the GSM. The subgrid pinning point parametrization in the GSM attempts to represent these features using a statistical scheme but mismatches with the PD ice shelf extent remain a challenge as discussed in other modelling studies (e.g. Albrecht et al., 2020b). Moreover, the coarseness of the model grid results in the marine core site being binned with the PD ice shelf grid cell. Without a proper accounting of structural error, model predictions at the marine core site might falsely

never deglaciate since the site is so proximal to a relatively stable ice shelf. Figure 5 shows the data-model score for paleoExt tier-1 data in AntICE2. Regardless, the full ensemble is able to capture the OMC in the region but the NROY simulations struggle to deglaciate the site. The ranges of the NROY sub-ensemble 2σ ice extent bracket the RAISED Consortium (2014) contours across



East Antarctica (Fig. 5). This is unsurprising given how few marine core observations exist across the East Antarctic continental shelf.

**(a) 20 ka**

**(b) 15 ka**

RAISED A Measured
RAISED A Inferred
RAISED B Measured
RAISED B Inferred

PD grounded ice
NROY subensemble +/- 2σ
NROY subensemble μ

**(c) 10 ka**

**(d) 5 ka**


**Figure 5:** The mean and 2σ range grounded ice extent for the not-ruled-out-yet (NROY) AN3sig sub-ensemble is shown by the black and dashed black line, respectively. It is compared against the RAISED consortium scenario A and B measured and inferred contours at a) 20 ka, b) 15 ka, c) 10 ka, and d) 5 ka. The 2σ ranges are the nominal 95% ensemble intervals based on the equivalent Gaussian quantiles.

In the Ross Sea sector, NROY simulations confidently bracket the paleoExt observations with the exception of two marine cores, which are closest to the continental shelf edge (2401, 2403). These PGL observations suggest an early retreat from the shelf edge, with the GL retreating over these sites around 27.5 to 23.9 ka. NROY simulations deglaciate later to remain consistent with the rest of the Ross Sea sector ice extent observations. The degrees of freedom in the ocean forcing can produce an initial partial retreat from the shelf edge since the full ensemble is able to capture these observations. However, a trade-off occurs between
capturing these continental shelf edge observations versus the remaining Ross Sea deglacial ice extent observations. When comparing the ranges or the NROY sub-ensemble ice extent to ice extent reconstructed by the RAISED Consortium in the Ross Sea



sector, the contours overlap broadly. The only exception is the western Ross GL at 15 ka where the simulated GL remains extended on the continental shelf for another few thousand years relative to the RAISED contours.

In the Amundsen Sea sector, the full ensemble and NROY sub-ensemble bracket the data quite well. However, areas with complex topography, small islands, and subgrid pinning points lead to misfits at core site 2502 for the NROY sub-ensemble. A series of marine sediment cores was taken along transects in several paleo-ice stream troughs, starting at the continental shelf edge and heading toward the coast. OMC were first recorded at the shelf edge as early as 19 ka (2508). However, other marine sediment cores from the outer to inner continental shelf, document a persistent early Holocene GL retreat starting at 12.5 ka until 7.9 ka (2511, 2513, 2514, 2516-2520). The full ensemble manages to bracket all but one OMC observation at 2520. The NROY ensemble manages to fit the past GL extent data along the Pine Island-Thwaites paleo-ice stream trough. However, NROY simulations struggle with the OMC data (251901 and 252001). In the Amundsen Sea sector, a persistent issue was the simulated PD ice shelf extent which would remain marginally too advanced, and which included smaller ice shelves coalescing to larger ice shelves as a result of the coarseness of the model resolution. This is attributed to resolution limitations of the ice sheet grid and ocean forcing, as well as the presence of subgrid pinning points that buttress the ice in the region. When comparing the ranges of the NROY sub-ensemble ice extent to the reconstructions by the RAISED Consortium (2014), the best-fitting sub-ensemble brackets the measured and inferred contours confidently. This includes observations that place the GL near the PD GL at Pine Island Bay at ~10 ka (Hillenbrand et al., 2013).

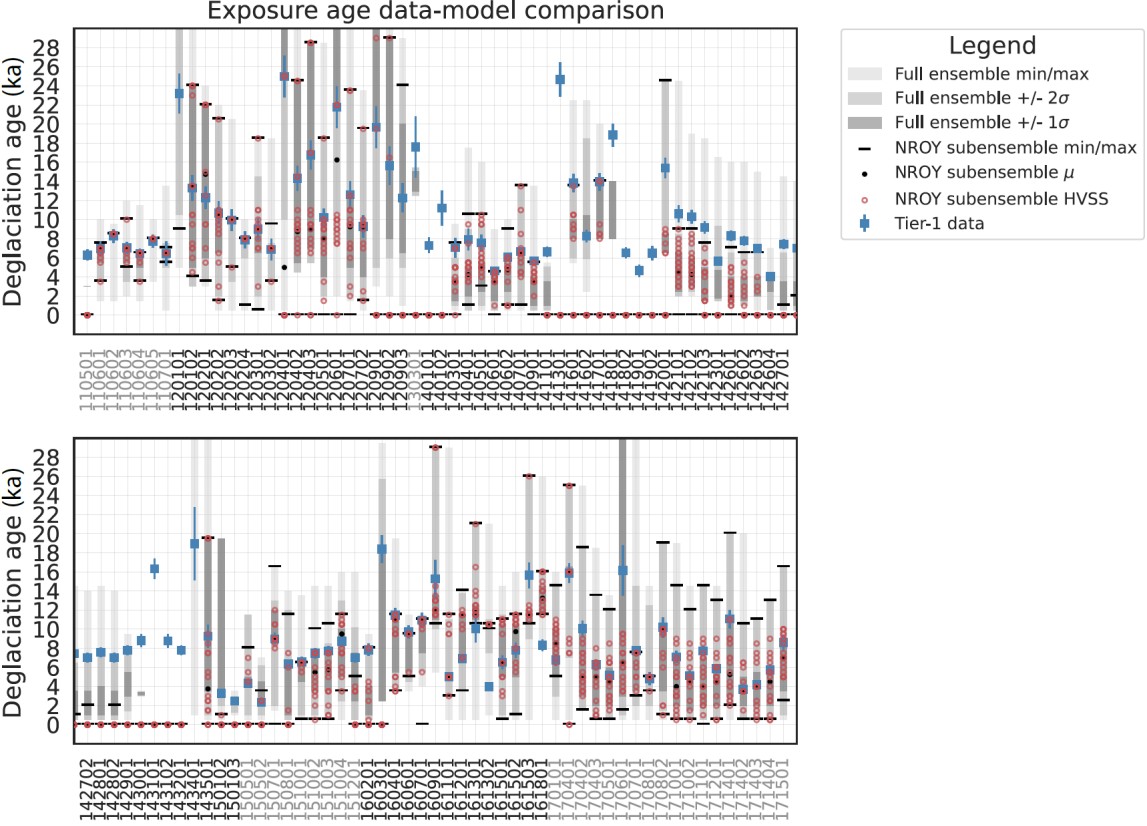

**Figure 6:** Past ice thickness data-model comparison for the highest quality tier-1 exposure data in AntICE2 at its respective elevation. The grey shading represents the min/max, 1 and 2σ ranges of the full ensemble. The solid black circles and lines are the mean and min/max ranges for the not-ruled-out-yet (NROY) AN3sig sub-ensemble. Simulations consisting of a high variance subset (HVSS) of the NROY AN3sig sub-ensemble are shown as red circles. The 2σ and 1σ ranges are the nominal 95% and 68% ensemble intervals based on the equivalent Gaussian quantiles, respectively. The AntICE2 paleoH data ID are shown in Figure S1.



The Antarctic Peninsula and Bellingshausen Sea sector is a topographically complex region with many features below the GSM resolution. During post-LGM deglaciation the GL retreated from 18.2 to 7.5 ka, albeit with significant regional variability. The full and NROY ensemble perform well in this sector given the aforementioned challenges, with a few exceptions. For example, there are two sites which are quite close to the coast which report a GL retreat at 9.2 ka (2609, 2610). While NROY simulations narrowly misfit 2609, not even the full ensemble brackets 2610. These sites are close to the coast and the basal topography was unfavour-

ably upscaled to produce a shallow marine environment and above sea-level topography, which resists deglaciation for the lack of direct ocean forcing. It is crucial to verify how the upscaling impacts the basal topography since some data-model comparison will be challenging without a proper accounting of such structural errors. The remaining reconstructions of post-LGM deglaciation based on marine sediment cores are captured by the full ensemble and NROY sub-ensemble, except site 2614 which is PGL at 11.8 ka. This core site is located near subgrid islands, potential pinning points, and PD grounded ice. These common challenges

occur frequently with the ice extent observations and explain the remaining misfits. With regards to the RAISED Consortium (2014) ice extent reconstructions, the GL ranges of the NROY ensemble bracket the measured contour in the Antarctic Peninsula-Bellingshausen Sea sector, except for the 10 ka contour. The AntICE2 data suggests the GL approached the PD coastline by 10 ka at many locations along the western Antarctic Peninsula shelf, as discussed above. This, however, conflicts directly with the RAISED Consortium (2014) inference at this time. Given the GSM is data-constrained by the AntICE2 database, a mismatch to the

RAISED contour is expected.

The Weddell Sea sector has few observations of past ice extent. The only marine core site for the shelf in front of the Ronne Ice Shelf (2701) consists of observations of OMC as early as 5.5 ka. The site is relatively close to the intersection of the bedrock above sea-level and the PD Ronne Ice Shelf margin. Therefore, it is unsurprising that the NROY simulations struggle at the site since overly extended ice shelves are a persistent challenge across the full ensemble. The remaining tier-1 observations near the Filch-

ner Ice Shelf front at core sites 2702, 2706, and 2708 document a PGL at 8.8, 1.9, and 12.9 ka BP, respectively and are bracketed by the NROY sub-ensemble. The RAISED Consortium (2014) proposed two distinct scenarios in the Weddell Sea sector, with scenario B being more compatible with recently published exposure ages from around the Weddell Sea embayment that propose much thicker ice upstream of the Ronne-Filchner ice shelf (Nichols et al., 2019). The NROY sub-ensemble ice extent contours bracket the RAISED Consortium (2014) scenarios for the Weddell Sea sector, particularly the measured extent.

At sites where the NROY sub-ensemble struggles to bracket the paleoExt observations (i.e. data-model score ≠ 0), the mismatch is usually caused by resolution limitations. There, poorly-resolved complex topography leads to mismatches between observed and simulated ice extent. This is particularly a challenge where subgrid pinning points can stabilize ice shelves or, similarly, where basal topography can stabilize the GL. This can impact the transient evolution of the ice margin which can yield persistent misfits that cannot be simply reconciled within the error model.

### 5.3 Past ice thickness

There are cosmogenic exposure ages taken from PD ice free regions scattered across Antarctica that constrain past ice thickness. The deglaciation age at its respective elevation (paleoH tier-1 data; Lecavalier et al., 2023), full ensemble (AN) statistics and NROY AN3sig sub-ensemble model prediction are shown in Fig. 6. The full ensemble and NROY sub-ensemble broadly bracket the paleoH observations with the exception of the Transantarctic Mountains. Instances, where the NROY simulations fail to capture

the observations, are discussed in the following.

Across East Antarctica, there are only two sites where the NROY sub-ensemble does not bracket the paleoH observations. A simulated deglaciation age of zero in Fig. 6 represents instances, where the site either never glaciated or never deglaciated. At both 1105 and 1303, the full ensemble manages to deglaciate the site but the NROY simulations fail to deglaciate those regions. This is a much broader issue in the Transantarctic Mountains, where 17 paleoH sites (e.g. 1401, 1402) are not bracketed by the

NROY sub-ensemble. At some of these sites, the full ensemble does manage to capture the exposure age constraints (e.g. 1416). However, the NROY simulations struggle to predict sufficient thinning in the Ross Sea sector. While in the Amundsen Sea, Antarctic Peninsula and Bellingshausen Sea, and Weddell Sea sectors, the NROY sub-ensemble brackets the paleoH data with the exception of five sites (1501, 1512, 1603, 1613, 1618). At these five sites, the full ensemble brackets the deglaciation ages, although the simulations responsible for this are ruled out, when considering the entire AntICE2 database.

Once more the NROY data-model misfits are attributed to resolution limitations. The 40 km by 40 km horizontal grid is based on upscaling the BedMachine version 2 subglacial topography, which effectively converts features such a nunataks and valleys that fall within a single grid cell into a uniform plateau. The fact that deep subglacial valleys are not resolved in topographically complex terrain has a considerable impact on ice dynamics. This manifests itself in entire regions excessively covered by thick ice because a region is simulated as a plateau and ice drainage is underestimated. This results in glaciated areas where ice is not

sufficiently thinning, and these misfits persist until the end of the simulation period (Fig. 7). The best examples of these regions are the Transantarctic Mountains, the Antarctic Peninsula, and Bellingshausen Sea sector. Moreover, by improperly resolving deep subglacial valleys, misattributions of the basal environment are possible (i.e. ice atop soft sedimentary substrate instead of



hard bedrock). The implementation of basal topography subgrid statistics in the basal drag scheme led to warm basal conditions
in subgrid valley glaciers, thinning ice in regions that tended to be too thick with respect to PD. Although it did not fully rectify
the excessive ice bias entirely, it improved the paleoH data-model misfits in certain regions. At some paleoH sites with excessive
PD ice in the model, the AIS thins in accordance with data constraints during the simulated post-LGM deglaciation. Thus, data-
model misfits of PD ice thickness do not necessarily imply an equivalent bias in the past.

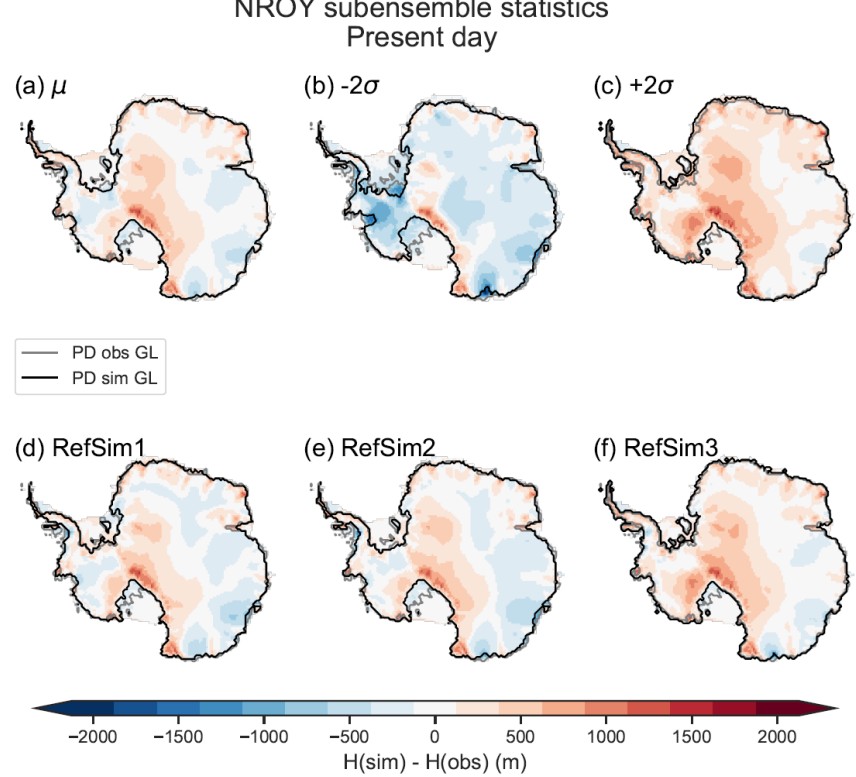

**Figure 7:** Present-day ice thickness data-model comparison for the not-ruled-out-yet (NROY) AN3sig sub-ensemble a) mean, b-c)
minus and plus 2σ, d-e) minus and plus 2σ. Three glaciology self-consistent simulations chosen from a NROY high variance subset
(HVSS): d) RefSim1; e) RefSim2; f) RefSim3. The 2σ ranges are the nominal 95% ensemble intervals based on the equivalent
Gaussian quantiles.

Mas E Braga et al. (2021) emphasized that the sampling position relative to the direction of ice flow can bias an exposure age.
This can result in significant paleoH data-model misfits when dealing with continental-scale ice sheet models since they do not
resolve a nunatak flank. The inability to resolve key features below the model horizontal grid size is a recurring theme to explain
patterns of data-model misfits in this analysis. However, resolving the nunatak flank is not within the scope of continental ice
sheet models, even of those operating at a computationally costly high spatial resolution with a 10 km by 10 km grid. Only models
that nest a domain around a nunatak or leverage adaptive grids may be capable of simulating the age offset caused by the
sampling location relative to mean flow. However, continental-scale AIS models with a constant horizontal grid resolution can
only hope to address this exposure age bias by broadening the error model and incorporating the mean flow direction relative
to the sample position. It has also to be taken into account that the mean flow direction is generally not reported alongside
exposure ages in the paleoH source studies.

### 5.4 Present day geometry
The PD geometry of the AIS is an essential boundary condition and a powerful constraint to evaluate model performance. Since
we are dealing with imperfect models operating at a relatively coarse model resolution, one would naturally expect misfits with
the PD observed geometry (Fig. 7). This constitutes the context by which to evaluate the performance of the GSM against PD
observations, particularly when comparing the PD misfits reported in this section to those of other studies (Seroussi et al., 2019a),




which solely focus on minimizing misfits to only the PD geometry using inverse approaches. As previously discussed, the aim is to avoid overfitting to the PD geometry by using an inversion scheme as to maximize the transient predictive capabilities of the model output.

The PD ice thickness misfit for the NROY AN3sig sub-ensemble is shown in Fig. 7. The NROY sub-ensemble mean is mostly ±250 m of the observations (Fig. 7a), which is reasonable given the model resolution and the uncertainties attributed to PD observed ice thickness across much of the ice sheet. The NROY simulations bracket the PD geometry observations as shown by the 2σ range of the NROY sub-ensemble (Fig. 7b-c). The NROY sub-ensemble minimum should exclusively demonstrate negative values while the maximum should demonstrate the converse. This is mostly the case across the AIS with some prominent exceptions. There are a few sites where the ice is too thin across the entire NROY sub-ensemble (blue areas shown in Fig. 7g), such as the Larsen C Ice Shelf, parts of East Antarctica, and ice in the Siple Coast region near the Ross Ice Shelf GL. In the case of the latter, the transient behaviour of the GL in the Ross Sea sector requires that it captures past ice extent/thickness observations and the PD GL position and geometry. This trade-off results in NROY simulations with a retreated GL in the Ross Sea sector and yields floating ice near the Siple Coast and, in turn, thinner grounded ice in the region.

The most prominent ice thickness misfits are found in the Transantarctic Mountains (Fig. 7b). As discussed in Section 5.3, the model resolution produces a flat bedrock plateau beneath the ice over much of the region rather than peaks with deep valley trough. This impedes ice flow and promotes the formation of a broad ice dome. Moreover, the subglacial substrate type is based on subgrid information from the BedMachine subglacial topography but ultimately, a threshold designates the ice in the grid cell as being underlain by either being unconsolidated sediment/till or hard bed. This favours hard bedrock basal conditions across much of the Transantarctic Mountains which again impedes ice discharge. Both characteristics are static in time, which suggests that the excessive PD ice thickness in the Transantarctic Mountains likely persist throughout the simulations.

When considering for the relative size of an ice shelf, the most impactful mismatches with the PD ice shelf extent affect small ice shelves which are at or marginally below the model grid resolution. Given the resolution of the model, some poorly resolved simulated ice shelves, such as those in the Amundsen Sea embayment and along the Bellingshausen Sea coast, manage to persist and buttress grounded ice. This can manifest in PD GL mismatches which can in turn lead to ice thickness misfits for the ice shelves and upstream of the GL. This also can affect larger ice shelves, for which discrepancies between the simulated and observed PD GL can produce considerable ice thickness misfits for the ice shelves.

## 5.5 Present day surface velocities

The ice flow velocity measurements for the AIS surface are based on observations taken from 2005 to 2017 (Mouginot et al., 2019). The surface ice velocity demonstrates fast moving regions, such as ice streams/shelves, and the slow-moving interior of the AIS. For this reason, the RMSEs are calculated for two regions delineated by a 2500 m elevation threshold. At locations in the AIS, where the PD surface elevation is greater than the threshold, usually slow-moving interior ice is present. Conversely, effectively faster-moving marginal ice is expected for areas below the surface elevation threshold. Such areas include ice streams/shelves. The two regions exhibit different sensitivities to parameter changes (basal ice deformation ensemble parameters), and, therefore, the scores were divided in two. For example, the interior surface ice velocity score is more sensitive to hard bed parameter choices when compared to the margin surface ice velocity score, which is very sensitive to the ice-shelf front and grounding line locations.

The spatial misfit with the PD surface ice velocities is shown in Fig. 8. The largest data-model misfits are observed at ice shelves and their tributaries. Any mismatch in PD ice shelf extent leads to large surface ice velocity discrepancies, such as in the Ross, Amundsen, and Weddell Sea sectors. If one excludes regions with mismatches in ice-shelf extent, the NROY simulations broadly bracket the observations within 2σ, especially when considering uncertainties affiliated with the observations (upwards of 5 m/yr). The NROY sub-ensemble 2σ surface velocities generally bracket the PD surface velocities of grounded ice (Fig. 8b-c). Any exceptions to this are associated with the ice shelves, specifically the Larsen C Ice Shelf and the Ross Ice Shelf, where modeled surface flow speed is either too slow or too fast. This can likely be attributed to the tributary glaciers or ice streams feeding these ice shelves and the potential misattribution of subglacial substrate type at crucial grid cells.




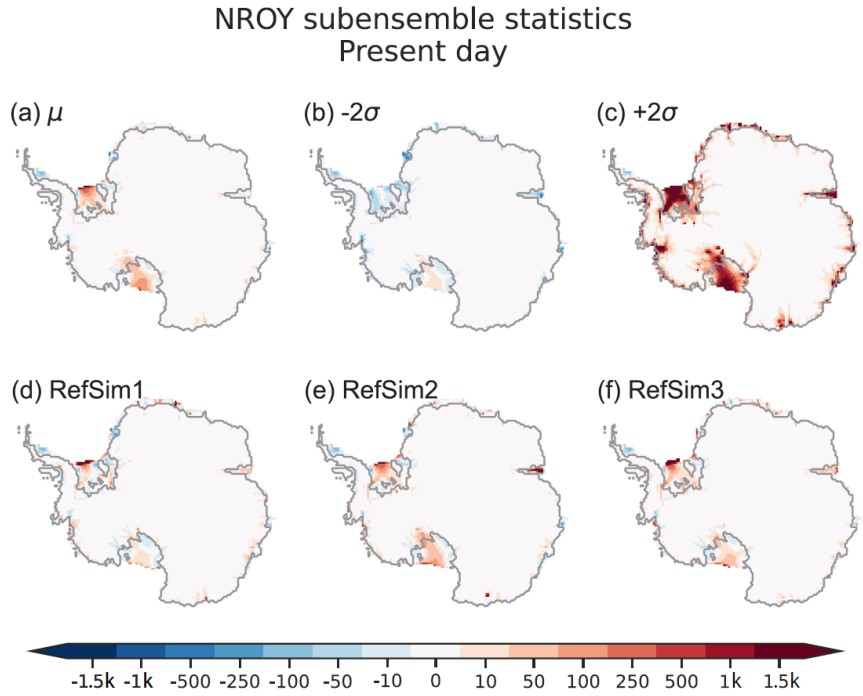

**Figure 8**: Present-day surface velocity model-data comparison for the not-ruled-out-yet (NROY) AN3sig sub-ensemble a) mean, b-c) minus and plus 2σ, d-e) minus and plus 2σ. Three glaciology self-consistent simulations chosen from a NROY high variance subset (HVSS): d) RefSim1; e) RefSim2; f) RefSim3. The 2σ ranges are the nominal 95% ensemble intervals based on the equivalent Gaussian quantiles.

## 6. Results

The AIS grounded ice volume for the full ensemble and progressively more data-constrained sub-ensembles are shown in Fig. 9. The full ensemble grounded ice volume demonstrates significant variance since the LIG. By history matching the ensemble, the grounded ice volume variance progressively decreases as the sieve becomes stricter from AN4sig to AN3sig. The 2σ and 1σ ensemble ranges shown across several figures (e.g. Fig. 9, Fig. 10, Table 3) are the nominal 95% and 68% ensemble intervals based on the equivalent Gaussian quantiles (2.275 - 97.725%, Gaussian 2σ quantiles and 15.866 - 84.134% Gaussian 1σ quantiles).

### 6.1 Last Interglacial

The LIG changes in grounded ice volume for the full ensemble and NROY sub-ensemble are displayed in Figure 9. At the termination of the penultimate glacial period (starting at ~135 ka) the AIS retreated rapidly, with its GL reaching a position upstream of its PD position in several AIS sectors during the LIG and thus contributing significantly to the LIG sea-level highstand (Fig. 10). Relative to PD, the AIS had a minimum grounded ice volume between -2.9 to -13.8 mESL as per the NROY sub-ensemble (Table 3). The AN3sig sub-ensemble presents a variety of LIG grounded ice deficit scenarios with ungrounding of marine-based sectors in the WAIS and/or EAIS. It should be noted that if marine-based grounded ice retreats, ocean water will flood the vacated submarine region. Therefore, only the ice above the point of flotation is initially responsible for sea-level rise as observed in far-field RSL records. In all instances, the AIS recovers relatively quickly after the LIG (~119 to 105 ka). Depending on the duration of the AIS LIG minima (Fig. 10d-f), it takes GIA rebound up to 10 kyr to raise the bed and displace ocean water away from Antarctic marine sectors. The viscous relaxation of the seafloor in formerly marine-based AIS sectors throughout the LIG therefore gradually increases the AIS contribution to far-field sea-level rise by displacing ocean water.



770

**Figure 9:** Antarctic grounded ice volume anomaly through time for the a) full ensemble AN; b) AN4sig sub-ensemble; and c) not-ruled-out-yet (NROY) AN3sig sub-ensemble. A high variance subset (HVSS) of the not-ruled-out-yet (NROY) AN3sig sub-ensemble is shown in grey which also includes three reference simulations (RefSim1, RefSim2, RefSim3) to illustrate glaciologically self-consistent simulation results. The 2σ and 1σ ranges are the nominal 95% and 68% ensemble intervals based on the equivalent Gaussian quantiles, respectively.

775





**Figure 10:** The histograms of key metrics are shown for the full ensemble (leftmost column), AN4sig sub-ensemble (middle column), and not-ruled-out-yet (NROY) AN3sig sub-ensemble (rightmost column). The key metrics of interest being a-c) the LIG
Antarctic Ice Sheet (AIS) grounded ice volume deficit relative to present day (PD); d-f) the timing of the LIG grounded volume minimum; g-i) the LGM grounded volume excess relative to PD; and j-l) the AIS contribution to MWP-1a. The 2σ and 1σ ranges are the nominal 95% and 68% ensemble intervals based on the equivalent Gaussian quantiles, respectively.

During the LIG sea-level highstand, GMSL has been inferred to 1.2 to 11.3 meters above present-day (Kopp et al., 2009b; Dutton et al., 2015; Düsterhus et al., 2016; Rohling et al., 2019; Dyer et al., 2021). For this period, the steric contribution was estimated at 0.8 m (Shackleton et al., 2020b; Turney et al., 2020), the glaciers and ice caps contribution was 0.32 ± 0.08 mESL (Marzeion et al., 2020). The Greenland Ice Sheet contribution to sea-level change during this period was constrained to 0.9 to 5.2 mESL (Tarasov et al., 2003; Dahl-Jensen et al., 2013; Goelzer et al., 2016; Yau et al., 2016; Bradley et al., 2018; Clark et al., 2020). A LIG sea-level highstand budget suggests a broad range of AIS contribution of -5.2 to 9.4 mESL. Therefore, the NROY AN3sig sub-ensemble AIS LIG ice deficit relative to present overlaps significantly with the LIG sea-level highstand budget. However, considering that



790    only ice loss from above floatation immediately contributes to GMSL rise, the AIS LIG sea-level contribution was reduced. The max LIG AIS volume deficit from AN3sig is 13.8 mESL (Fig. 10 and 11). Fig. 12 illustrates the source regions which underwent the greatest amount of ice loss during the LIG in the NROY AN3sig sub-ensemble. The bulk of the mass loss is across the West Antarctica, with a retreated GL with respect to PD in the Ross, Amundsen, Bellingshausen and Weddell Sea sectors. Many regions across the WAIS experience ice-sheet thinning in excess of 1000 m. The NROY sub-ensemble suggests that in limited areas the

795    EAIS was a few hundred meters thinner relative to PD, particularly, in the Wilkes–Victoria Land sectors. With George V Land being the only EAIS sector with a GL significantly landward of the PD position. Three simulations from the HVSS are shown in Fig. 12d-f to illustrate the variety of configurations that yield distinct LIG configurations. Fig. 12d illustrates a partially collapsed WAIS (mainly ungrounding of the Thwaites Glacier and Siple Coast ice stream drainage basins) with a seaway connecting the Amundsen and Ross Sea sectors, while Fig. 12e shows a nearly full WAIS collapse with seaways connecting the Weddell, Bellingshausen,

800    Amundsen, and Ross Sea sectors. Fig. 12f demonstrates a fully collapsed WAIS and a more pronounced retreated grounded ice margin in Victoria Land.

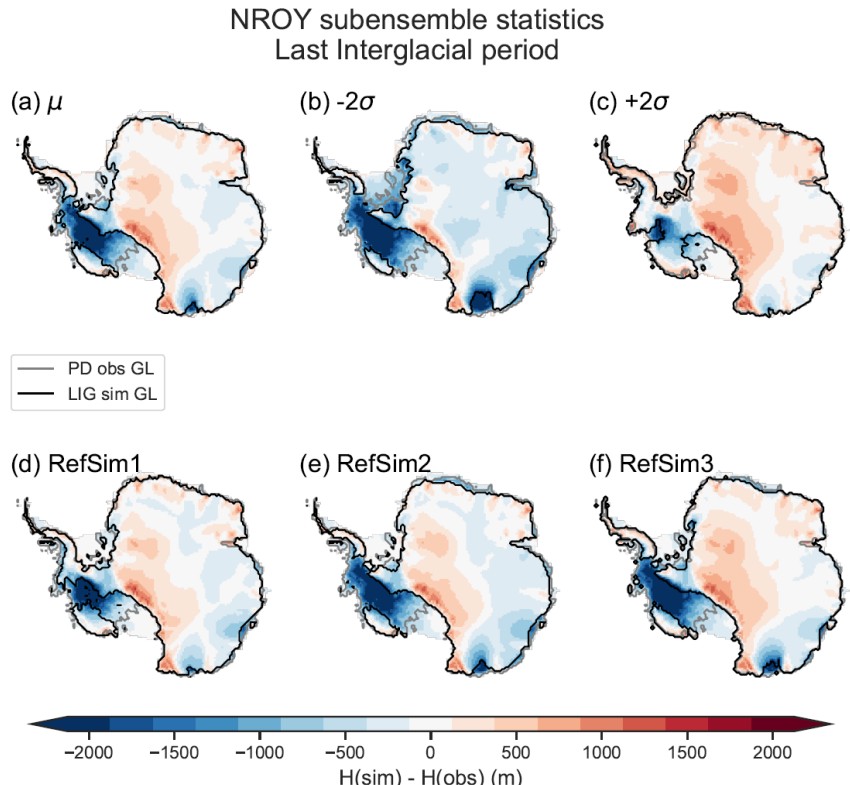

**Figure 12:** The not-ruled-out-yet (NROY) AN3sig sub-ensemble a) mean and b-c) 2σ range are shown during the LIG. Three glaciology self-consistent simulations chosen from a NROY high variance subset (HVSS) are showcased: d) RefSim1; e) RefSim2; f)

805    RefSim3. The 2σ ranges are the nominal 95% ensemble intervals based on the equivalent Gaussian quantiles.

The main caveat to the LIG AIS simulation results remains the lack of observational constraints during the LIG. This translates to a large variance across the NROY sub-ensemble. Due to the lack of constraining records during this key period of interest, the model parameters which induce the greatest sensitivity for the AIS LIG sea-level contribution remain poorly constrained (Fig. S3 to S7, S10). Therefore, it is difficult to rule out either a low-end or a high-end AIS LIG sea-level contribution from the NROY sub-

810    ensemble, given the considerable impact of poorly constrained parametric uncertainties and very limited data-constraints. Moreover, the extent and strength of sub-surface ocean warming at the ice sheet margin during the LIG remains highly uncertain. This implies that no definitive statements can be made regarding the closure of the sea-level budget without undertaking a history-matching analysis in the future, when novel constraints specifically targeting the LIG climate forcing and ice geometry, if and whenever any become available.



## 6.2 Last Glacial Maximum

During the LGM the NROY AN3sig sub-ensemble has min/max grounded ice volumes of 9.2 to 26.5 mESL excess relative to PD (Table 3). The AntICE2 database mostly consists of data spanning the post-LGM deglaciation, and as the observations were more strictly imposed on the full ensemble during the history-matching analysis from 4σ to 3σ thresholds (Fig. 10 and Table S1), the overall variance decreased, and smaller LGM ice volumes were sieved out. In the AN4sig sub-ensemble, there remained AIS simulations with an LGM excess volume of 6.3 mESL; by imposing a 3σ sieve threshold, the AN3sig sub-ensemble ruled out these smaller LGM excess volumes (Table 3).

Fig. 13 shows the NROY AN3sig sub-ensemble mean and 2σ range LGM ice thickness difference relative to PD and LGM GL position which illustrates where more ice than at PD was stored during the LGM. Previous studies typically yielded smaller AIS LGM volumes between 5.9 to 14.1 mESL (e.g. Whitehouse et al., 2012b; Argus et al., 2014; Briggs et al., 2014; Albrecht et al., 2020b). But critically, none of these studies explicitly showed that their model had the degrees of freedom to produce larger AIS configurations and evaluated their inconsistency with data constraints. The LGM GL advanced to the continental shelf edge in most sectors. Towards the interior of the EAIS, certain regions were thinner during the LGM relative to present due to reduced precipitation, which agrees with earlier modelling studies (Golledge et al., 2012). This is particularly illustrated when evaluating a single glaciologically self-consistent simulation (Fig. 13d-f). The sectors most responsible for the LGM ice excess are shown in Fig. 11. In these sectors the PD GL is very far away from the continental shelf edge but had advanced to near the shelf edge at the LGM. Thus, significantly more ice could be stored on the shelf there, hence the larger LGM contributions from the Ross and Weddell Sea sectors.

The main differentiating factors between the largest versus smallest LGM reconstructions in the NROY sub-ensemble (26.5 vs 9.2 mESL) are the GL extent on the continental shelf, and the ice surface slope towards the interior. The latter can be attributed to parameter choices yielding a till basal drag and climate forcing conducive to thicker ice to build up and persist (Fig. S10). Moreover, it requires basal conditions with basal stresses and drag that are capable of supporting thicker ice. The ice thickness on the shelf impedes the ability of the ice sheet interior from easily displacing ice to the margin where it is more susceptible to negative mass balance over the course of the glacial cycle.

When the AIS reaches its LGM extent, it decreases the total area of the Earth's ocean by $3.5 \times 10^{12}$ m$^2$ (1% decrease). This represents a relatively modest decrease in the global ocean area. However, for a present-day ocean area of $3.618 \times 10^{14}$ m$^2$ it marginally decreases the water equivalent ice volume needed to produce a 1 m GMSL change. When discussing ice sheet sea-level contributions, it is important to explicitly state whether it is in relation to a dynamically changing ocean area or entirely referenced to the PD ocean area. The GIA model accounts for migrating shorelines but the mESL estimates presented in this study are derived on the PD ocean surface area.

## 6.3 Deglaciation

The post-LGM deglaciation represents the period during which the model is heavily data-constrained by AntICE2. The NROY AN3sig sub-ensemble simulations illustrate the timing of the local LGM at 15.7 ka (Fig. 11). The deglaciation begins gradually and peak rates of mass loss are not simulated until 10.7 ka. In all instances, the NROY AN3sig simulations all provide a very minor AIS contribution to MWP1a from -0.2 to 0.3 mESL (minimum and maximum contributions). The history matched NROY simulations provide a considerable constraint on the AIS contribution to MWP1a. When compared to the source region contributions as inferred by far-field RSL observations (0 to 5.9 mESL from the AIS; Lin et al. (2021)), it illustrates that near-field observations rule out a significant MWP1a sea-level contribution from Antarctica. Moreover, the rate of mass loss from the AIS over the MWP1a interval is not anomalous to the background rate of mass loss during the deglaciation. This implies the AIS did not contribute towards an acceleration in sea-level rise during the MWP1a period. GIA model simulations focused on far-field RSL observations and AIS simulations data-constrained by near-field observations (AntICE2) provide a consistent and conclusive result, MWP1a is clearly not sourced from the AIS.

Over the course of the deglaciation, the AIS retreated most dramatically from 12 to 4 ka (Fig. 11, S11 and S12). This includes major grounding line retreat across the continental shelf occurring during the early to middle Holocene. Many sectors reach their present-day extents around ~4 ka. In particular based on the NROY AN3sig sub-ensemble mean, the GL in the the Ross Sea sector had retreated upstream of the PD ice-shelf front by 6 ka to reach the PD GL by 4 ka. In the Amundsen sector, the GL retreated in a series of steps across the continental shelf over the course of the Holocene with the most prominent retreats occurring between 12-10 ka and 6-4 ka. Around the Antarctic Peninsula and on the Bellingshausen Sea shelf, the majority of marine-based ice retreat occurred from 16-10 ka, when the PD ice margin is reached at many locations. In the Weddell Sea embayment, grounded ice reached the PD Ronne-Filchner ice shelf fronts around 10 ka, and the PD GL position was reached by 4 ka. The Dronning Maud Land and Victoria Land sectors are characterized today by narrow continental shelves where grounded ice reached the PD GL between 6-4 ka. In Prydz Bay (= Amery Ice Shelf sector) grounded ice was present at the continental shelf edge until 12 ka, when it started to retreat to reach the PD GL by 6-4 ka. The timeline described above is based on the NROY AN3sig sub-ensemble mean



chronology, and a wide variety of chronologies are described within the NROY sub-ensemble, enabling a multitude of distinct timing and regional retreat scenarios from a more modest LGM extent.

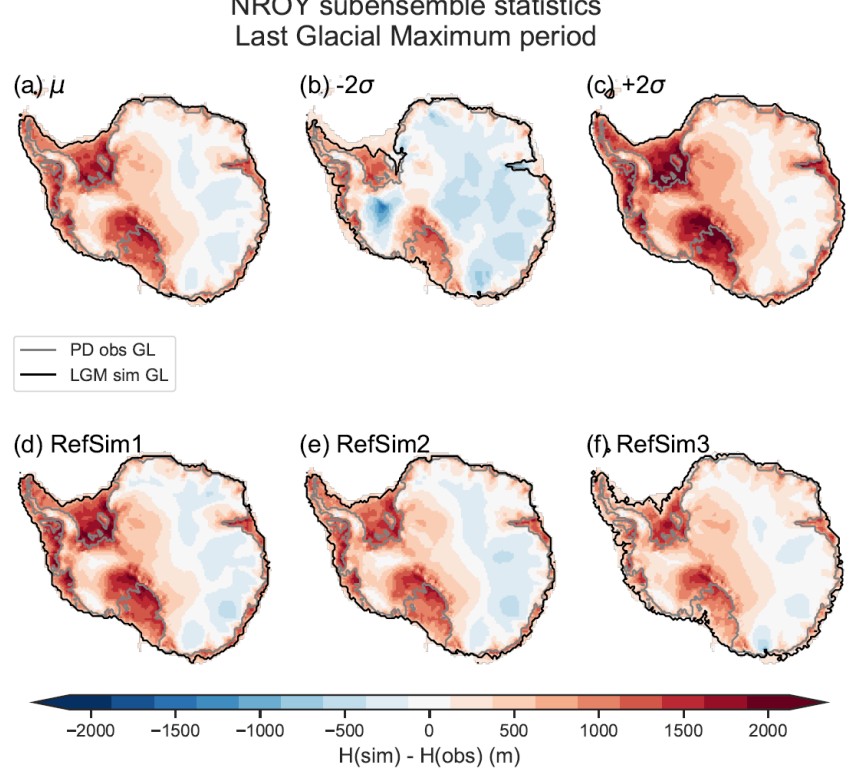

**Figure 13:** The not-ruled-out-yet (NROY) AN3sig sub-ensemble a) mean and b-c) 2σ range are shown during the LGM. Three glaciology self-consistent simulations chosen from a NROY high variance subset (HVSS) are showcased: d) RefSim1; e) RefSim2; f) RefSim3. The 2σ ranges are the nominal 95% ensemble intervals based on the equivalent Gaussian quantiles.

Over the Holocene, a few studies have discussed the viability of a GL retreat landward of its PD position. This was reported in the eastern Ross Sea sector, where subglacial sediment cores taken across the Siple Coast retrieved sediments which had carbon ages dating back to approximately the LGM (>20 ka), implying a retreated GL during the early Holocene, as the inferred ages were discounted as the actual timing of GL retreat (Kingslake et al., 2018). Other studies have indicated that these ages are inconsistent with other observations and offered alternative interpretations, suggesting maximum GL retreat during the middle or late Holocene (Neuhaus et al., 2021; Venturelli et al., 2023). Due to their ambiguous interpretation, these data were not included in the AntICE2 database as constraints. Some of the best-fitting AN3sig sub-ensemble simulations tend to yield a re-treated GL position with respect to PD in the Ross Sea sector during the late Holocene, but these simulations do not reconstruct GL re-advance in time to match the PD GL position. The climate forcing and its degrees of freedom were unable to yield a sufficient GL retreat in the Ross Sea sector during the last deglaciation followed by a re-advance towards the PD position. It is possible, however, that the climate forcing envelope in the model inadequately represents the appropriate regional forcing to enable a re-advance. Alternately, the radiocarbon ages of the subglacial sediments from the Siple Coast sector may need to be reinterpreted (cf. Neuhaus et al., 2021; Venturelli et al., 2023).

## 6.4 Present-day AIS

At PD the AIS is in a non-steady state. The transient evolution of the AIS implies that our ability to understand the present and future state of the AIS is contingent on its past trajectory. Model simulations that investigate the transient evolution of the AIS at present and in the future tend to spin up their ice sheet models (e.g. Golledge et al., 2015b; DeConto and Pollard, 2016b; Albrecht et al., 2020b). Alternatively, some studies initialize models using data-assimilation approaches which presume the PD observations as an accurate steady-state representation of the AIS (Cornford et al., 2015; Fürst et al., 2016; Pattyn, 2017). The



latter approach achieves simulations with the smallest RMSE to the PD geometry (Seroussi et al., 2019a), but offer limited pre-
dictive capabilities given the risk posed by overfitting to PD observations (Schannwell et al., 2020). Therefore, paleo spin-up
approaches are much better suited to evaluate the transient evolution of the AIS and the full breadth of systemic sensitivities.

The aim of a transient model spin up is to retrace the thermo-mechanical trajectory of the ice sheet over time to properly initialize
the thermal memory of the system and basal environment in preparation for exploratory experiments (e.g. paleo simulations or
future projections). By prioritizing the transient behaviour of the system, paleo spin-up initializations usually lead to larger PD
misfits as compared to data assimilated initializations (Seroussi et al., 2019b). The resulting PD bias can be used to correct model
predictions and subsume their bias into error model. In future projections, a paleo spin-up preserves the sensitivity of the ice
sheet due to past warm and cold periods. The paleo model calibration and spin-up conveniently constrain the parameter space
and encapsulate all past uncertainties into the PD boundary conditions for potential AIS projections. Our best-fitting NROY AN3sig
sub-ensemble results represent a series of paleo spin-up boundary conditions which can be employed as initialization conditions
to evaluate PD and future AIS changes. Moreover, they can be used as a basis to propagate uncertainty bounds forward in time
to help quantify projection uncertainties.

## 7. Conclusion

This study represents a history-matching analysis of AIS evolution since the last interglacial. This was achieved through a history-
matching analysis, where a truly large ensemble of simulations (N=9,293) was constrained by a comprehensive observational
database (AntICE2; Lecavalier et al., 2023). Simulations were considered NROY by the data, if the simulations were within 3σ of
the highest quality data in the AntICE2 database (tier-1 and 2 data). This yielded a NROY sub-ensemble termed AN3sig, which
comprises of 82 simulations. The NROY sub-ensemble exhibits a wide range of viable reconstructions and represents bounds on
the evolution of the AIS during past warm and cold periods.

The configuration of the AIS during the LIG lacks near-field observational constraints and its modeled reconstruction depends on
an uncertain oceanic forcing. The NROY sub-ensemble yields a grounded ice volume deficit relative to present of 2.9 to 13.8 mESL.
These wide bounds are predominantly the product of parametric uncertainties associated with sub-surface ocean temperatures
for the LIG. Conversely, the configuration of the AIS during the LGM and the post-LGM deglaciation is better constrained by the
AntICE2 database. During the LGM, the AIS had an excess grounded ice volume of 9.2 to 26.5 mESL relative to present. This raises
the possibility that the LGM AIS was significantly larger than previously thought. The history-matching analysis over the last glacial
cycle yields a variety of viable AIS changes that enable a more meaningful evaluation of the atmospheric/oceanic circulation and
sea-level budget during the LIG and LGM. Future research will focus on addressing remaining data-model misfits that are not
bracketed by the full ensemble and NROY sub-ensemble, improving the representation of structural uncertainties in the error
model, and achieving probabilistically robust model predictions as outlined in Tarasov and Goldstein (2021).

## Author Contribution

B.S.L. and L.T. led and designed the study. B.S.L. wrote the manuscript with editorial input from co-author. B.S.L. and L.T. ran the
model simulations, processed the dataset and simulation output. B.S.L. and L.T. performed the data-model analysis, and L.T.
conducted the majority of the BANN training, and MCMC sampling. B.S.L. visualized the model results.

## Competing interests

The authors declare that they have no conflict of interest.

## Acknowledgements

We thank Greg Balco Claus-Dieter Hillenbrand, and Christo Buizert for discussions and helpful reviews on a draft. This text also
benefited from reviewer comments. Support provided by Canadian Foundation for Innovation, the National Science and Engi-
neering Research Council, and ACEnet. This research has been supported by an NSERC Discovery Grant held by Lev Tarasov
(number RGPIN-2018-06658) and is a contribution to the PalMod project funded by the German Federal Ministry of Education
and Research (BMBF).




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
