# Peer review of "A history-matching analysis of the Antarctic Ice Sheet since the last interglacial – Part 1: Ice sheet evolution"

_EGUsphere, 2024_

## Author Comment (AC1)

We would like to thank the reviewers for their referee comments, suggestions, and feedback. This response aims to address any comments raised by the reviewers. Our responses are embedded below and are shown in orange.

**Response to referee comments #1**

Summary

The work of Lecavalier & Tarasov uses a new numerical model, the Glacial Systems Model (GSM), to reconstruct the evolution of the Antarctic Ice Sheet (AIS) since the Last Interglacial through a very large ensemble of simulations constrained by likely the most comprehensive paleo-ice sheet constraint database up to date (AntICE2). The GSM is not just an ice sheet model but presents coupled modules and parameterisations allowing it to consider several different processes that affect ice sheet dynamics. As a result, a total of 38 model parameters can be varied throughout the ensemble to assess different types of uncertainty inherent to (paleo) ice sheet modelling. The parameter space covered for each parameter was refined through what the authors call "waves", i.e., different iterations of their large ensemble run, which were then used to train an emulator that, in combination with MCMC sampling, helped determine which range of values the model was most sensitive to, and at the same time produced results that best matched the AntICE2 constraints within the standard deviation boundaries allowed by the authors. Once the final, most refined, ensemble of runs was obtained, the AIS evolution was assessed for key periods: the Last Glacial Maximum and Meltwater Pulse 1A. A proper evaluation of the Last Interglacial was hindered by the lack of suitable paleo constraints.

Apart from refinements in terms of assessing the AIS volume at the LGM and its contribution to Meltwater Pulse 1A, I believe this manuscript's greatest contribution is the presentation of a novel and much more comprehensive methodology for constraining ice sheet history from large ensemble modelling, and a numerical model that integrates several processes in a way that allows for an unprecedented number of (uncertain) parameters to be explored. For this reason, both these key points need to be described very clearly and able to be reproduced, which is the main point where this manuscript needs improvement.

This work definitely brings enough innovation to merit publication and will be of wide interest to the numerical modelling and (paleo)glaciology communities, but there are substantial issues which need to be addressed before I can recommend it to be published. Apart from a reorganisation of the introduction and some rewriting or clarification of some sections, for which my comments and suggestions are listed below, the model-description paper cited as *Tarasov et al., in prep* needs to be available in some way for a proper appreciation of this work by both reviewers and readers. I assume, since it is "in prep", that there is no pre-print available. I expect that a publicly available pre-print would provide enough information to solve this main issue, both for me as a reviewer and for the potential readers that will come across this paper before the model description is published in its final version. As mentioned, my comments regarding this paper are provided below, since I believe the study merits being published eventually (and to expedite the process), but not before details on GSM are available, and that the results presented here can be reproducible.

**General comments**

**1.**My greatest concern with the manuscript is the fact that it uses an unpublished model, for which there is no available manuscript (referred to as "Tarasov et al., in prep") or reference to "data and code availability". This poses a serious issue for reproducibility and for a proper assessment of several parts of the methodology, as I believe these are not in this manuscript because they will be properly described in the cited in-prep manuscript. Considering the model presents several improvements and more complex parameterisations/couplings than the current ice sheet models, including its older version (Briggs et al., 2013; TC), e.g., in basal sliding dynamics, sub-grid pinning point parameterisations, different surface boundary conditions, basal melting of floating and grounded ice, it is of extreme importance that proper descriptions of all these innovations are available. Similarly, the way the three forcing schemes are described (L243) is rather vague, and without a thorough description, it is hard to assess how they differ from each other. In a similar fashion, the way how the data-model scoring is done should be more explicit, as it is hard to evaluate from the descriptions given for each data type. Without seeing the in-prep manuscript, I believe this fits better in this manuscript than in the in-prep paper (L370).

The GSM description paper will be submitted next Monday to GMDD and a copy of the preprint will be made available on Lev Tarasov's website https://www.physics.mun.ca/~lev/". This should address the model related issues that the reviewer raises.

Additional responses to the comment above:

It should be stated that Briggs et al., 2013 does not use the GSM but rather the Penn State University ice sheet model (PSU-ISM).

There are a variety of approaches to data-model scoring (e.g. Briggs et al., 2013; Ely et al., 2019) and the one applied in this study is broadly described in Tarasov & Goldstein (2019). Therefore, another ice sheet modeller could leverage their ice sheet model of choice to achieve a history-matching analysis. This manuscript is already exceeding lengthy, hence why we rely on citations and opted to exclude an exhaustive model description and data scoring methodology section. Furthermore, the details of the history matching implementation is a whole paper on its own and is currently in preparation.

**2.**The description of model initialisation/spinup is not clear. There are some hints of part of it throughout section 3, e.g., it seems to start from PD geometry, but there are no details on how the internal ice thermal structure was initialised, or how the spatial distribution of basal drag was obtained apart that it depends on elevation. The description given in L291-302 is not enough to properly evaluate how this was done. I assume a better description will be available in the in-prep paper, but this is not available for a proper understanding and fair assessment of the methodology.

More information was added regarding model initialization.

"The Antarctic simulations were initialized at 205 ka using the PD AIS geometry. The englacial temperature was initialized using an analytical approximation of the EDC ice core borehole tempature profile. The basal ice is scaled to a temperature below the pressure melting point to stabilize the initial ice dynamics. The initial ice velocities are computed using a shallow ice approximation solution over a 1.5 kyr period to achieve a partial thermal equilibrated initialization

prior to transient hybrid ice physics calculations. The model is spun up to the penultimate glacial maximum at ~140 ka to minimize any dependencies on the initalization." L340 – L345

The basal drag scheme is described in the Tarasov et al., preprint and the till sediment distribution which specifies the basal drag coefficient ranges are further detailed in Lecavalier, 2024.

Lecavalier, B.S. (2024). A history-matching analysis of Antarctic Ice Sheet evolution since the last interglacial (Doctoral dissertation, Memorial University of Newfoundland).

**3.** The assessment of Section 6 is hindered due to the fact that Fig. 11 is completely absent from the manuscript. From the text, I have the feeling that this figure is key to understand the discrepancies between the much larger ice volume simulated during the LGM – an important piece of this work.

When the manuscript package was compiled for submission, this figure was unintentially left out and this has been corrected. The figure was renamed to figure 13 and demonstrates which sectors stored the most ice duing the LGM. L905

**4.** The introduction needs a thorough rearrangement of paragraphs. The content is good, it indeed introduces all concepts and sets up the problem that the paper wants to address. However, it does so in a rather erratic way, and there is little connection between the paragraphs. For example, I would expect the paragraph starting on L47 (which sets up the problem and paper aims) to come at the end of the introduction. Other concepts that need to be introduced, such as the key periods targeted by the paper (MWP1a, LIG, LGM), data-related issues (types of data for ice sheet reconstructions, their availability, and their use in modelling), and outstanding research questions have been presented, but all come after what reads as the closing introduction paragraph. Similarly, current model limitations (paragraph at L36) should come much later in the introduction, after discussing the points mentioned above. Therefore, I strongly recommend the contents of the introduction to be revised and rearranged, with some rewriting to better link paragraphs and improve the flow of reading.

As per your recommendations, the introduction was rearranged and more connecting sentences were added to improve the flow across the introduction. L25-L172

**5.** Why did the authors opt to use TraCE-21ka to reconstruct their forcing? Considering the transient signal is reconstructed with a glacial index, instead of using TraCE throughout, this choice needs further justification.

The transient TraCE-21ka ocean forcing is applied throughout its available (21ka to 0ka) time period. The glacial index scheme is applied for older time periods. The text was revised to clarify this point at L269-L272:

"This calculates mass balance at the ice front, beneath the ice shelves, and at the grounding line. The ocean temperature forcing is based on transient TraCE-21ka simulations (He, 2011) which are PD bias corrected by the Estimating the Circulation and Climate of the Ocean (ECCO) reanalysis ocean temperatures (Fukumori et al., 2018). For ocean forcing temperatures going back beyond 21 ka, the glacial index scheme is applied to the PD bias corrected TraCE-21ka predictions."

Fukumori, I., Heimbach, P., Ponte, R. M., & Wunsch, C. (2018). A dynamically consistent, multivariable ocean climatology. *Bulletin of the American Meteorological Society, 99*(10), 2107-2128.

**6.** It would be worth expanding briefly on how the classification of data as tier1 to tier3 is done (suggestion: a summarised version of the 1st paragraph of section 3.1 in Lecavalier et al., 2023; ESSD). Even though the methodology is detailed in the original paper, given the strong relevance to the assessment performed in this manuscript, it is worth expanding a bit more on it.

These details are succinctly found in Lecavalier et al., 2023, it comes down to extracting the data with the highest constraining power on past AIS change. The text was revised to provide an example of tier-1 versus tier-2 data but any additional detail would only bloat an already long paper.

"The ice sheet simulation is then scored against the data described in AntICE2, with a predominant focus on tier-1 and 2 data. Tier-1 data is the highest quality data which has the greatest power to constrain the ice sheet and GIA model (e.g. exposure age data constraining LGM ice thickness). While tier-2 data supplements tier-1 by providing more granular detail on past changes (e.g. exposure age data constraining the deglacial timing and thinning rate). Tier-3 data is excluded from the history-matching analysis since it correlates highly with the higher quality tier-1/2 data and is mostly used for visual comparison (Lecavalier et al., 2023)." L359-L364

Lecavalier, B.S., Tarasov, L., Balco, G., Spector, P., Hillenbrand, C.D., Buizert, C., Ritz, C., Leduc-Leballeur, M., Mulvaney, R., Whitehouse, P.L. and Bentley, M.J., 2022. Antarctic Ice Sheet paleo-constraint database. Earth System Science Data Discussions, 2022, pp.1-34.

**7.** The concept of "waves" should be more clearly described (likely in the paragraph starting in L382), because the amount of work put into such an "ensemble of ensembles" really cannot be properly appreciated until much further down the manuscript.

The text has been revised to more clearly define the "waves of ensembles" conducted in this study at L399–L404:

"As part of this study, several large-ensemble data-constrained analyses were iteratively performed to evaluate the model's ability to bracket AntICE2. A series of large-ensemble model simulations were performed iteratively, where a given iteration constitutes a wave of simulations consisting of anywhere between 500 to 2000+ simulations. GSM simulation output was applied towards supervised machine learning of Bayesian Artificial Neural Networks (BANNs) for Markov Chain Monte Carlo (MCMC) sampling to efficiently explore relevant portions of the parameter space. A flow chart is shown in Figure S8 that illustrates the history-matching algorithm and the waves of large-ensemble simulations conducted in this study."

**Specific comments**

L196: How are the ice flow approximations coupled in the hybrid scheme? Are they just superimposed, or is there a (e.g., weighted) transition between them?

The GSM uses the dynamical core of the Penn State University Ice Sheet Model (Pollard & DeConto., 2009, 2012; DeConto & Pollard, 2016) as cited in the model description. The hybrid SIA-SSA scheme use superpositions of the depth-integrated SIA and SSA equations.

L242: The mention of the glacial index here does not read well. You first mention about the three forcing schemes, then mention the glacial index, and then proceed to present the forcing schemes.

I suggest mentioning the glacial index first, and that it is used by (seemingly) two of the three schemes, before describing them.

The glacial index scheme is linked to the climate forcing by providing temporal evolution to spatial climate reconstructions, hence why it is described alongside the climate forcing description. A glacial index scheme is a common method applied in ice sheet models studying past ice sheet changes. The text was revised to add more detail:

"The glacial index scheme uses a glacial index derived from the EPICA Deuterium record ($\delta D = \delta^2 H$) (EPICA, 2004; Jouzel et al., 2007). The glacial index provides temperoral evolution to spatial reconstructions. The glacial index is effectively a temperature anomaly relative to present which is normalized such that the LGM is equal to one (e.g. Tarasov and Peltier, 2004; Niu et al., 2019)." L250-L253

Niu, L. U., Lohmann, G., Hinck, S., Gowan, E. J., & Krebs-Kanzow, U. T. A. (2019). The sensitivity of Northern Hemisphere ice sheets to atmospheric forcing during the last glacial cycle using PMIP3 models. *Journal of Glaciology*, *65*(252), 645–661. doi:10.1017/jog.2019.42

L243: The three forcing schemes could benefit from a more thorough description that would help in evaluating and reproducing the results.

A general overview is provided on the climate forcing scheme to demonstrate what is dinstinct from other studies. A more comprehensive description is found in the Tarasov et al., preprint.

L251: Please make it clear whether the positive-degree-day and positive-temperature-insolation surface-melt schemes are used for all forcing schemes or not.

The positive-degree-day and positive-temperature-insolation scheme are used to calculate surface melt for all three climate forcing schemes.

L264: What is the reasoning for grouping Dronning Maud Land with Wilkes and Victoria Land? The former is rather far apart from the latter two and is under a different oceanographic regime. This needs to be either properly justified or they need to be separated.

From the PD bias corrected TraCE-21ka ocean forcing, the 3D ocean temperature field is extrapolated under the ice shelves. Ocean marine sectors were only specified to  provide regional bias shifting ensemble parameters to enable adequate bracketing of the past and present GL and ice shelf extent. For much of the East Antarctic coast this was not necessary, the ocean temperature climatologies correlate across large parts of the EAIS, especially for regions with a limited continental shelf like Dronning Maud Land and Wilkes-Victoria Land. This is the primary reason why the Lambert-Amery sector was excluded from the previous grouping since ocean temperatures beneath the Amery ice shelf are sensitive to a different ocean temperature profile given how ocean waters are advected beneath this large ice shelf.

L271: It is not clear how the far-field global sea-level forcing affects the ice sheet in GSM. I assume through their solid-Earth model, and that it would be better described in the model-description paper, but this needs some sort of description here.

The text was revised to provide more context on the sea-level forcing. This is more clearly discussed in the accompanying Antarctic GIA paper (Lecavalier et al., In prep).

"Within the GSM, the benthic stack and RSL observations drive the far-field global sea-level forcing (Lambeck et al., 2014b) when performing joint ice sheet and GIA calculations. After a transient AIS simulation finishes, the AIS chronology is amalgamated into the GLAC3 global ice chronology to perform fully gravitationally self-consisten sea-level calculations." L279-L281

L273: This paragraph could use some rewriting. Not only do the sentences not properly link well, but it is not clear how the basal forcing conditions are computed. Furthermore, it is not clear how the two different fields are blended. Is the "ensemble parameter" some sort of weight between two fields? This is only made explicit way further down, in L525, and should also be clearly stated when first describing the method. Finally, the first sentence could be better formulated: wouldn't basal topography and thickness already yield ice-surface elevation? Or do the authors mean "bedrock topography"? I can see this being argued and explained in different ways (i.e., among bedrock topography, ice-base topography, ice thickness, and ice-surface topography), but as it stands it is not consistent.

Basal topography and ice thickess generally yields the ice surface elevation but this is not the case for floating ice shelves so specificity is required. The text was revised to improve clarity and provide additional detail (L282). Technical details on the model description can be found in the Tarasov et al., preprint.

L355: Please clearly describe how "a quadrature was calculated".

It is simply the square root of the sum of the squares. The text was revised the explicated say this to avoid confusion. "The square root of the sum of the squares is calculated of all the borehole temperature profiles to obtain a borehole temperature profile score for a given simulation." L372

L396: This paragraph is very helpful in explaining the types of uncertainties that have to be dealt with in the history-matching approach. It should be moved further up, before or when the term "structural uncertainty" is used for the first time (Section 4.1).

The text was revised accordingly.

L415: I find it problematic to use a method not fully described anywhere that is deferred to a future publication. At least some level of detail should be given, even if it is an appendix to the paper.

There are a variety of approaches to data-model scoring (e.g. Briggs et al., 2013; Ely et al., 2019) and the one applied in this study is described in Tarasov & Goldstein (2019). Therefore, another ice sheet modeller could leverage their ice sheet model of choice to achieve a history-matching analysis. And as previously mentioned, this topic is a whole paper in itself that will soon be submitted for publication.

L444: This sentence could be rewritten in a much clearer way. Suggestion: "Simulations beyond the thresholds for each data type (Table S1) were then ruled out from the history-matching analysis (i.e., sieved out)."

The text was revised with:

"Simulations beyond the implausibility thresholds for any data type (Table S1) were then ruled out as part of the history-matching analysis" L463

L508-513: I can follow the reasoning, but the writing feels a bit disconnected. It would be good to rewrite. For example, stating that you are talking about vertical advection – this might not be clear considering the mention of "higher velocities at warm beds drawing in ice", which I believe is a reference to horizontal movement.

"When evaluating the best-fitting NROY sub-ensemble, the temperatures of type 1 profiles tend to remain clustered relatively close to the observations. Conversely, the NROY sub-ensemble results at type 2 profiles show significant variance. Simulations that produce cold englacial temperatures achieve this because colder ice from higher in adjacent ice columns is advected in." L524-L527

L539: Why are the ice stream profiles relegated to a lower tier? I had understood that data tiers were linked to their quality, but this phrasing suggests it is by how well the data is captured by the ensemble. Perhaps it is just bad wording, but please clarify.

The Siple Coast borehole profiles from Siple Dome, Bindschadler, Kamb, and Alley/Whillans ice streams are relatively proximal and correlate with each other, so they are assigned to a tier-2 status with the Siple Dome profile, however, remaining the regional tier-1 representative (Lecavalier et al., 2023). Including them all in the tier-1 category does not improve the constraining power of the data significantly.

L550: I am not sure I followed the reasoning in this sentence. Do you mean that the subsequent "ensemble waves" aimed to improve this mismatch? Or what kind of analysis/prioritisation was done? Or do you mean that you will focus on the more inland temperature profiles? Please clarify.

We are not quite sure what the source of the reviewer's misunderstanding is. Additional large ensembles of simulations are done to explore other promising portions of the parameter space. This can result in previously mismatched data becoming bracketed by newer simulations. We have tried to improve the clarity of this section, more clearly defined what is meant by "waves" and added additional context:

"Data-model comparisons shown in this section can illustrate instances where the full ensemble or NROY sub-ensemble fail to bracket the observations, however this does not necessarily imply the simulations are entirely inconsistent with the data given structural uncertainties." L506-L508

"The full ensemble brackets the observed profiles at the Law Dome, Talos Dome, Skytrain Ice Rise, and Berkner Island, albeit not by the 2σ range. The NROY sub-ensemble fails to bracket the observations at these borehole sites. Additionally, at the Fletcher Promontory, the simulated temperatures are far too warm at the base. Considering these borehole sites are surrounded by complex basal topography that is poorly resolved, the analysis prioritized to capture the temperature profiles in the interior of the ice sheet." L569-L573

L808: The figures mentioned are quite heavy in information, and it is not trivial to find exactly which parameters the authors refer to from a glance. It would be helpful to name which parameters are considered to induce the greatest sensitivity instead of just referring to the figures.

The Figures S3-S7 show historgrams for the full ensemble and NROY sub-ensemble for all the model parameters to show how their distribution changes after imposing the implausibility thresholds. While Figure S10 shows a correlation heatmap of the model metrics/scores against the ensemble parameter. The metric "volgLIGdiff" shown in these supplementary figures is the grounded ice volume different between the LIG and PD. VolgLIGdiff correlates with the model parameter rTOceanWrm which is poorly constrained. Moreover, the metric volgLIGdiff and rTOceanWrm has a high variance with both the full ensemble and sub-ensemble. The text was revised to specify this.

"Due to the lack of constraining records during this key period of interest, the model parameters (e.g. rTOceanWrm) which induce the greatest sensitivity for the AIS LIG sea-level contribution (volgLIGdiff) remain poorly constrained (Fig. S3 to S7, S10)." L835-L837

Some of the abbreviated metrics in Fig. S3 to S7, S10 were not defined in the text and the figure captions were revised to include their definitions.

"The ensemble parameters are defined in Table 1, the individual model scores are defined in Table S1. The present-day (PD) metrics shown are the PD grounded ice volume (volg0), PD floating ice volume (volf0), PD West AIS grounded ice volume (volgWAIS), PD East AIS grounded ice volume (volgEAIS), PD grounded ice area (arg0), PD floating ice area (arf0), PD West AIS grounded ice area (argWAIS), and PD East AIS grounded ice area (argEAIS). The LGM metrics shown are the 20 ka grounded ice volume (volg20), 20 ka grounded ice volume excess relative to present (volg20diff), 20 ka grounded ice area (arg20), 20 ka grounded ice area excess relative to present (arg20diff). The Meltwater Pulse 1a (MWP1a) metric is the grounded ice volume change over the MWP1interval (volgMWP1a). The last interglacial (LIG) metrics are the timing of the LIG AIS minimum (timeLIGmin), LIG grounded ice volume deficit relative to present, and LIG grounded ice area deficit relative to present.".

L813-814: This is a very important statement and should be used as an opportunity to guide further field efforts rather than just acknowledge a limitation to hopefully be addressed in the future. I think the authors should explicitly state/suggest what kind of "novel constraints" would be helpful, and which regions should be prioritised for data collection (e.g., in which data-poor region was the ensemble range the widest?).

Looking at the +/- 2sigma range in figure 11,12, S11, and S12 wherever the ranges are the largest illustrate regions that are poorly constrained by the data given the uncertainties in the entire glacial system. The text was revised to emphasize this result (L956-L957).

L835: Please mention which parameter combinations you refer to when mentioning Fig. S10, as it is not trivial to hunt for different parameter pairs in such a large heatmap.

Any parameter with a correlation greater than 0.25 and less than -0.25 with volg20diff (grounded ice volume difference between 20 ka and present day) demonstrates which parameters show a relationship with LGM ice volume excess (fnpre – glacial index scaling coef for precipitation, rlps – LGM temperature lapse rate, POWbtill – soft bed power law exponent, rHhp0 – grounding line parametrization selection, earthUV – upper mantle viscosity). L867

**Technical corrections**

L99: there is an extra '(' in the citations.

Corrected.

L109: Greenland Ice Sheet**s**? It is not clear why it is in plural.

Corrected.

L120: I don't think the paper DOI should be listed in the citation.

Corrected.

L184: Terms such as "paleoExt" and "paleoH" should be clearly defined at their first mention. This only happens in L363 and L357, respectively.

Corrected.

L267: TRACE here is all capitalised, please change for consistency.

Corrected.

L274: "Poorly observed", as there is no hyphenation for adverbs

Corrected.

L278: "an envelope of *model* realisations", for clarity

Corrected.

L286: either "across Antarctica" or "across the Antarctic"

Corrected.

L345: I believe a comma separating these two sentences works better than a full stop

Corrected.

L483: "metrics of interest"

Corrected.

L513: Please remove the comma

Corrected.

L551: temperature profiles

Corrected.

L659-660: There should be no commas in this sentence

Corrected.

L728: Please remove "ice shelves", as this is already clear from how the sentence begins.

Corrected.

L732: This sentence needs rewriting for better readability. Suggestion: "Slow-moving ice is usually present at inland locations where the PD elevation is above this threshold."

Text was revised.

L733-735: These two sentences can be merged. Suggestion: "Conversely, faster-moving marginal ice is expected for areas below the surface elevation threshold, which include the aforementioned ice streams/shelves."

Text was revised.

L855: Please replace the comma for "that", and "is" for "was"

Corrected.

L907: "This study presents"

Corrected.

L909: Please remove the comma.

Corrected.

L930: There's a comma missing between "Balco" and "Claus"

Corrected.

**Figures**

Figure 1: Please add subpanels where the main locations mentioned in the text are marked. I appreciate that adding all names to the already existing figure would not work, as there would be too much clutter, hence the suggestion to add the most important regions discussed as subpanels.

The named places in the text are mostly all labeled on the figure with a few exceptions. For the few places not labeled on the figure, their location is either well known in the community or associated with a region in the text. For example, Pine Island Bay being mentioned in relation to a broader discussion about the Amundsen sector.

Figure 10: Please make the x axes have the same maxima and minima for each row, so the different ensembles can be more easily compared, and increase the font size of the legend in each plot.

The trade off when increasing the font size in the legends is that it decreases the size of the historgram plots. In the final submission, this figure is in a vector format which improves legibility significantly over the rasterized format in the preprint. Given the effort involved, we will await guidance from Copernicus technical staff about final format edits for the figures.

Figure S1: Please reorder the panels according to the order that they are mentioned in the text and refer to the respective subpanel when mentioning Fig. S1 in the main text. Please increase the figure resolution for figures S2, S11, S12.

In the final submission, these figures will be uploaded in a vector format which preserves detail and clarity rather than the rasterized format in the preprint. The order they are shown is based on the data

type ID: past ice thickness = 1, past ice extent = 2, borehole temperature profile = 5, GPS uplift rate = 8, past RSL = 9.

**Response to referee comments #2**

**General comments**

In this paper, the authors summarized the results of nearly 30,000 ensemble experiments based on the dataset established by the authors, reconstructing the history of the Antarctic Ice Sheet from the Last Interglacial to the present. The papers about the model itself and GIA are published separately, so it is difficult to make a judgement on these. But, I believe this series of studies will make a significant contribution to this research field.  After all related papers published, a concise summary paper would make a greater contribution to the community.

While this paper provides a detailed discussion of the data and model, one concern is that the novelty of this paper is not clear. Although it addressed the missing ice problem of the LGM, the wide range of AIS volume makes it difficult to conclude that it provides a solution. In the abstract, the authors only mention the LGM AIS, but I think that they should also mention the LIG and MWP1A.

The novelty of this work is with the history-matching analysis of the AIS which demonstrates the possible history of the AIS that is consistent to a large and diverse observational constraint database. This work represents the largest assessment of uncertainties in the Antarctic glacial system and carefully considers both model and data uncertainties. By carefully considering the uncertainties in the glacial system, we demonstrate the degree by which we fundamentally can understand past AIS evolution. This yields results (mininum and maximum bounds) that are methodologically more robust than fine tuning and small ensemble studies.

The primary result are those pertaining to the AIS LGM and deglaciation since this period is relatively well constrained by the AntICE2 database. The AIS contribution to MWP1a are stated in the abstract. The AIS MWP1a contribution are minimal across the full ensemble and NROY sub-ensemble which reinforces the growing consensus that the AIS contributed very little to this interval. The AIS during the LIG are also mentioned in the abstract but primarily to emphasize the considerable uncertainties during this period, especially given the uncertainties in the climate forcing and the lack of observational constraints. As we clearly state, our analysis provides a potential solution. Yes the uncertainties are large, but that is because we've more fully assessed them than previous studies.

**Specific and technical comments**

- The term "resolution limitation" appears frequently, but there should be mention of the spatiotemporal resolution of the model in the Methods section.

Resolution limitation always refers to the horizontal resolution limitations, this has been explicitly added to the text. The vertical resolution is not a limiting factor as compared to the horizontal resolution and proves adequate for such studies. The temporal resolution is subannual and is reduced if a given time-step fails to converge. A comment on the adaptive temeral resolution has been added to the text (L339): "The ice dynamics temporal resolution is annual to subannual, it is adaptively reduced whenever ice dynamic calculations fail to converge."

- L68: In Fig. 1, it is not clear whether it is marine-based.

A choice was made here to not show the modern day contour where the basal topography equals zero to delineate marine-based sectors to avoid making the figure any busier. This is commonly shown across countless Antarctic publications and this contour is only accurate for the present-day AIS geometry and bedrock elevation.

- L69: mESL → Spell out as it is the first occurrence in the main text.

Revised.

- Figure 2: The label of LIG is hard to see. Since bars do not clearly indicate the period, shades like those used for the Holocene and Glaciation should be used.

Label on plot was relocated to improve clarity.

- L120: DOI information is unnecessary.

This was an reference manager import issue, this and other referencing issues has been corrected.

- L152-L155: Deschamps et al. (2012) is cited in consecutive sentences, and different durations for MWP1A seem to be mentioned. Please check the citation.

The first mention is a summary of several studies, while the second mention specifically reports on Deschamps et al., 2012. The text was revised to improve clarity (L132-L141).

- L187: Spelling out GSM is unnecessary.

Revised.

- L527-529: Need reference.

References were added but the following statement was part of an initial assessment when specifying prior parameter ranges for the geothermal heat flux blending parameter.

- L542: resolution limitaton → What is the temporal resolution of this model?

The GSM uses a subannual adaptive temporal resolution, this has been added to the text as per mentioned in a previous response.

- L553: What is the BC?

Spelled out the boundary condition acronym.

- L557: facto red - > factored

This typo is not present in my version, could be an artefact when the pdf was built.

- Similarly for Figure 6, annotations are needed in Figure 4 because it is not clear which site numbers correspond to which areas.

The site IDs are shown in Figure S1 and the different site IDs are shown in grey or black to delineate the main Antarctic sectors. This is also detailed in Lecavalier et al., 2023. Some text was added to guide the reader on how to more easily interpret the site IDs by sectors:

"The first digit of a site ID or datapoint ID is associated to the data type (past ice thickness (paleoH) = 1, past ice extent (paleoExt) = 2, borehole temperature profile = 5, GPS uplift rate = 8, past RSL = 9), while the second digit is associated to the sector (Dronning Maud-Enderby Land = 1, Lambert-Amery = 2, Wilkes-Victoria Land = 3, Ross Sea = 4, Amundsen Sea and Bellingshausen Sea = 5, Antarctic Peninsula = 6, Weddell Sea = 7; sector boundaries are shown in Fig. 1)." L183-L187

- ex. L595, 605, and 609: When interpreting using data numbers, the relevant previous studies should be cited.

We've tried adding a few references where there is a focussed discussion (or is data already cited). But to go beyond this would entail citing all data used in the AntICE2 database. This would imply that every modeller who uses a database for constraint would have to cite every study (in our case hundreds) contributing the relevant data. This is not standard practice for obvious reasons. The issue of adequate academic credit for data providers is a challenge that still lacks a feasible solution. The challenge even becomes more clear if one considers a reanalysis data-product using data from thousands of weather stations.

- The relationship between the data numbers and ice extent is unclear. An enlarged figure to clarify this relationship would be helpful. L648: It is mentioned to be consistent with RAISED, but does this apply to both Scenario A and B?

Figure S1 shows the extent data IDs. The RAISED consortium only has two scenarios in the Weddell sea sector. Given the latest exposure data from the Weddell sector, the community is favouring scenario B with its extended GL margin during the LGM. This is explicted stated in the text and a revision was made to clarify this further. L669

- Figure 10: Hard to see the legend. Since the legend is common in the figure, the bigger legend can put somewhere in the figure.

The legend is different for each frame since it includes statistics on each individual histogram. When final figures are uploaded, these figures will be in a vector format rather than a rasterized format which will improve clarity.

- The discussion on the missing ice from the LGM should be included in section 6.2.

The text was expanded to discuss the importance of these large AIS configurations during the LGM with regards with the LGM sea-level budget/missing ice problem. However, without running global GIA simulations using a variety of global ice chronologies and Earth models, conclusive answers on the missing ice problem and any associated discussions is limited. The text was revised to comment on this in section 6.2:

"The larger AIS geometries in the NROY sub-ensemble can considerably contribute towards closing the sea-level budget and resolving the missing ice problem. Albeit some of the LGM excess ice is grounded below sea-level which partially negates the Antarctic contribution to a sea-level lowstand during the LGM. Moreover, to conclusively quantify the contribution of the AIS to the missing ice problem based on far-field RSL observations, additional GIA simulations are required using a variety of global ice chronologies and Earth models. The accompanying paper, Lecavalier et al., (In prep),

discusses these research objectives and future modelling is planned to quantify the AIS sea-level contribution to past global sea-level change." L876-L882

- In the discussion of deglaciation, while the amplitude of sea-level fluctuations is certainly important, the temporal resolution of the model also needs to be considered. Does this model have sufficient temporal resolution to discuss MWP1A? To detect the start and end times of a 500-year MWP1A, a temporal resolution of several hundred years is necessary.

As addressed in another comment above, the ice dynamics temporal resolution is less than one year and can be adaptively reduced to smaller time steps when required. Therefore, the model is well suited to study the MWP1a interval.

Thank you for giving the opportunity to review this paper.

---

## Referee Report (RR1)

After having read the revised manuscript, I must say its quality and readability have drastically improved. Combined with the revisions, having access to the model description pre-print was very helpful in addressing the main questions related to technical aspects of the model and was sufficient for appropriately conducting this review process. Whereas I am not a reviewer of the modelling description paper, nor will I make any comments on the manuscript content itself, I could not help but get stuck on the author's statement *"Depending on community interest and involvement, a github for the model may eventually be setup."*. I rather believe that, if the model is not available in a standard way like GitHub, with some documentation, it will be much harder to get traction in the community. This is of course just an encouragement from my end for the authors to seek to expand the group of users of their glacial systems model, which this paper highlights to be extremely useful for the paleo ice sheet modelling community.

My only remaining concern about this manuscript is regarding some of the citations. As per TC's manuscript submission guidelines, citing a pre-print posted on a personal webpage is not allowed for publication (i.e., no "in prep", "in review", or "submitted" references are allowed, only those that were given a DOI). Thus, the pre-print should be available in GMD Discussions with a citeable DOI before I can recommend this manuscript to be published. I have been checking GMD's website daily to make sure I did not miss it being out until the last day I could delay sending my review, but so far it has not appeared. If the manuscript has already been submitted and is waiting on an Editor to open the discussion, I would suggest liaising with GMD so its process can be expedited and a citable GMDD version of the model description can be made available. Similarly, it would be highly beneficial if the accompanying "Part 2" paper would be available for the reader as a pre-print, so its citation can be kept. On a similar note, I could not find the reference "Tarasov & Goldstein (2019)" as per the author's rebuttal letter. Assuming it was meant to be "Tarasov & Goldstein (2021)", in GMDD, I am unsure what TC's policy is regarding citing pre-prints that did not have their revised version accepted. This is beyond my role as a reviewer, but I found it to me my job to highlight this to the Editor and the authors.

Citation problems apart, I am happy that the introduction is much improved and the reading flows really well throughout the entire manuscript despite its considerable length. I think this manuscript is a good contribution, showcasing the power of data constraints in paleo ice sheet models. My remaining points are rather minor, mostly editorial.

L10: Are citations in the abstract allowed? It should not be necessary, and the mention of AntICE2 (including its full name) should be sufficient

L90: I think there's an "If" missing at the beginning of the sentence

L132: "can quantify" reads better and is more appropriate for the middle of an introduction than "will quantify"

L168-171: not quite framed as "research questions". It might be worth rewording or calling them "research problems/goals".

L240: the reference Morlighem et al. (2024; SciAdv, doi:10.1126/sciadv.ado7794) would be good to support your approach to treat ice-cliff failure in a more conservative way

L251: "temperoral" -> temporal

L253-255: Does the first scheme then just adds an uniform anomaly based on the glacial index, which is further modified by the lapse rate? Please clarify

L255: just "van Wessem et al."

L372: Please define that this sum is the quadrature, so you can appropriately use the term in L378

L385: It might be worth reiterating that the evaluation of the mentioned consequences is presented in the "part II" paper

L438: the proper way of writing isotopes is $^{14}$C and $^{10}$Be

L500-505: It is not clear what the criteria were for picking those 18 runs for the HVSS, neither how exactly RefSims 1 to 3 were deemed to "collectively represent the best-fitting simulation (or simulations?)". Do you just mean the top 3 performing simulations when evaluating their score against the data? Please clarify

L508: "necessarely" -> necessarily

L523-524: Whereas I think I understand this sentence after reading it 3 times, it could be rewritten for better readability

L526: It might be worth saying "larger spread" or something similar instead since, as far as I understand, you are not showing their variance, nor testing whether it is statistically significant. Whereas I do not personally think this is a problem, I can see some readers getting stuck in this sentence because they expected something different to be shown/discussed

L538: either "on a magnetic and a seismic inference" or "on magnetic and seismic inferences" would read better

L544: I would start the sentence with "Particularly" to avoid repetition (two instances of "especially" are four words apart), and change "are" for "is", unless "overlap" was meant to be plural

L589: I believe an "and" is missing before "open"

L677: This sentence feels odd, as it implies that this is not the norm. It could be rephrased as "Cosmogenic exposure ages taken from PD ice free regions scattered across Antarctica (can) constrain past ice thickness"

L820: "across West Antarctica", without "the"

L839: "data constraints", no need for hyphenation

L869: I'd emphasise that you are still talking about the "continental shelf" here, even if it is not strictly necessary

Table 1: Is it correct that the first two EOFs are used for temperature, but only the first for precipitation?

Figure 2: please explain in the caption to what the lighter blue shading refers

Figure 3: should it be "full ensemble statistics" or "full ensemble spread" in the caption?

Figures 4 and 6: There is no explanation why some sites have their ID in grey or black. Please add it to their caption.

Figure S9: Is there any particular reason why the HVSS markers are grey in the plot, but black in the legend?

---

## Author Response (AR2)

We would like to thank the reviewer for their comments, suggestions, and feedback. This response aims to address any comments raised by the reviewer. Our responses are embedded below and are shown in orange.

**Response to referee comments #1**

**General comments**

After having read the revised manuscript, I must say its quality and readability have drastically improved. Combined with the revisions, having access to the model description pre-print was very helpful in addressing the main questions related to technical aspects of the model and was sufficient for appropriately conducting this review process. Whereas I am not a reviewer of the modelling description paper, nor will I make any comments on the manuscript content itself, I could not help but get stuck on the author's statement *"Depending on community interest and involvement, a github for the model may eventually be setup."*. I rather believe that, if the model is not available in a standard way like GitHub, with some documentation, it will be much harder to get traction in the community. This is of course just an encouragement from my end for the authors to seek to expand the group of users of their glacial systems model, which this paper highlights to be extremely useful for the paleo ice sheet modelling community.

This is the intention of this upcoming publication, to detail all of the GSM's components, adverstize it, make it publically available to other research groups, and with an expanded community, build out version control and comprehensive documentation.

My only remaining concern about this manuscript is regarding some of the citations. As per TC's manuscript submission guidelines, citing a pre-print posted on a personal webpage is not allowed for publication (i.e., no "in prep", "in review", or "submitted" references are allowed, only those that were given a DOI). Thus, the pre-print should be available in GMD Discussions with a citeable DOI before I can recommend this manuscript to be published. I have been checking GMD's website daily to make sure I did not miss it being out until the last day I could delay sending my review, but so far it has not appeared. If the manuscript has already been submitted and is waiting on an Editor to open the discussion, I would suggest liaising with GMD so its process can be expedited and a citable GMDD version of the model description can be made available. Similarly, it would be highly beneficial if the accompanying "Part 2" paper would be available for the reader as a pre-print, so its citation can be kept. On a similar note, I could not find the reference "Tarasov & Goldstein (2019)" as per the author's rebuttal letter. Assuming it was meant to be "Tarasov & Goldstein (2021)", in GMDD, I am unsure what TC's policy is regarding citing pre-prints that did not have their revised version accepted. This is beyond my role as a reviewer, but I found it to me my job to highlight this to the Editor and the authors.

The GSM description paper was submitted on Sept 20[th] 2024, a pre-print should be made available shortly upon making the code available with a permanent DOI. A pre-print for part 2 can be found at: https://doi.org/10.5194/egusphere-2024-3268. The Tarasov & Goldstein (2021) study can be found at: https://doi.org/10.5194/cp-2021-145.

Citation problems apart, I am happy that the introduction is much improved and the reading flows really well throughout the entire manuscript despite its considerable length. I think this manuscript is a good contribution, showcasing the power of data constraints in paleo ice sheet models. My remaining points are rather minor, mostly editorial.

**Specific comments**

L10: Are citations in the abstract allowed? It should not be necessary, and the mention of AntICE2 (including its full name) should be sufficient

Addressed.

L90: I think there's an "If" missing at the beginning of the sentence

Addressed.

L132: "can quantify" reads better and is more appropriate for the middle of an introduction than "will quantify"

Addressed.

L168-171: not quite framed as "research questions". It might be worth rewording or calling them "research problems/goals".

Addressed.

L240: the reference Morlighem et al. (2024; SciAdv, doi:10.1126/sciadv.ado7794) would be good to support your approach to treat ice-cliff failure in a more conservative way

Added.

L251: "temperoral" -> temporal

Corrected.

L253-255: Does the first scheme then just adds an uniform anomaly based on the glacial index, which is further modified by the lapse rate? Please clarify

Effectively yes, a scalar glacial index forcing plus 10°C forcing per $pCO_2$ doubling and lapse rate vertical temperature adjustment is applied to the PD climatology (GSM descrption paper; Tarasov et al., submitted to GMD). This is a similar scheme found in Pollard and DeConto, (2012). This is one of the several schemes applied to produce a wide envelop of possible climate scenarios.

L255: just "van Wessem et al."

Corrected.

L372: Please define that this sum is the quadrature, so you can appropriately use the term in L378

Addressed.

L385: It might be worth reiterating that the evaluation of the mentioned consequences is presented in the "part II" paper

Addressed.

L438: the proper way of writing isotopes is 14C and 10Be

Corrected.

L500-505: It is not clear what the criteria were for picking those 18 runs for the HVSS, neither how exactly RefSims 1 to 3 were deemed to "collectively represent the best-fitting simulation (or simulations?)". Do you just mean the top 3 performing simulations when evaluating their score against the data? Please clarify

Added:

"The simulations that make up the HVSS were selected based on maximizing the normalized multi-dimensional distance between metrics and scores for simulations in the NROY sub-ensemble. A few reference simulations with minimized scores for key data types were also included in the HVSS, such as the best overall scoring simulation, best scoring simulation against ice core data, and best scoring simulation to marine paleo extent data."

L508: "necessarely" -> necessarily

Corrected.

L523-524: Whereas I think I understand this sentence after reading it 3 times, it could be rewritten for better readability

"Ice dynamics then perturb the temprature profile of the ice by displacing colder ice from the surface deeper into the ice column"

L526: It might be worth saying "larger spread" or something similar instead since, as far as I understand, you are not showing their variance, nor testing whether it is statistically significant. Whereas I do not personally think this is a problem, I can see some readers getting stuck in this sentence because they expected something different to be shown/discussed

Addressed.

L538: either "on a magnetic and a seismic inference" or "on magnetic and seismic inferences" would read better

Corrected.

L544: I would start the sentence with "Particularly" to avoid repetition (two instances of "especially" are four words apart), and change "are" for "is", unless "overlap" was meant to be plural

Addressed.

L589: I believe an "and" is missing before "open"

Corrected.

L677: This sentence feels odd, as it implies that this is not the norm. It could be rephrased as "Cosmogenic exposure ages taken from PD ice free regions scattered across Antarctica (can) constrain past ice thickness"

Corrected.

L820: "across West Antarctica", without "the"

Corrected.

L839: "data constraints", no need for hyphenation

Corrected.

L869: I'd emphasise that you are still talking about the "continental shelf" here, even if it is not strictly necessary

Addressed.

Table 1: Is it correct that the first two EOFs are used for temperature, but only the first for precipitation?

Yes, a sensitivity analysis showed that the second precipitation EOF had effectively no impact on the simulation output. For this reason it was dropped as an ensemble parameter.

Figure 2: please explain in the caption to what the lighter blue shading refers

Caption was revised. It is showing the raw and gaussian filtered record.

Figure 3: should it be "full ensemble statistics" or "full ensemble spread" in the caption?

Revised the caption to simply say: "…where the grey shading represents the min/max, 1σ and 2σ ranges of the full ensemble."

Figures 4 and 6: There is no explanation why some sites have their ID in grey or black. Please add it to their caption.

It is to more easily delineate data in different Antarctic sectors. The caption was revised to include: "The data ID transitition in colour to demarcate the data located in different Antarctic sectors."

Figure S9: Is there any particular reason why the HVSS markers are grey in the plot, but black in the legend?

Updated the figure legend to be consistent with the figure.